# FIRST TRY MATTERS: REVISITING THE ROLE OF REFLECTION IN REASONING MODELS

## ABSTRACT

Large language models have recently demonstrated significant gains in reasoning ability, often attributed to their capacity to generate longer chains of thought and engage in reflective reasoning. However, the contribution of reflections to performance improvement remains unclear. In this paper, we systematically analyze the rollouts of eight reasoning models on five mathematical datasets. We focus on reflective behaviours where the model has already produced an answer but continues reflecting before finalizing its output. Our analysis reveals that reflections are predominantly confirmatory and rarely alter the model's initial answer, a pattern consistent across models and datasets. To understand the role of reflections in training, we construct supervised fine-tuning (SFT) datasets with varying amounts of reflection steps. We observe that training models on rollouts with more reflection steps primarily enhances first-answer correctness rather than the ability to correct initially wrong answers through reflections. This motivates us to propose a question-aware early-stopping method that enhances inference-time token efficiency by stopping the reasoning process once a few plausible candidate answers are generated, thereby reducing unnecessary reflection steps. Motivated by this, we further propose to dynamically truncate the reflections after a candidate answer has appeared during generation, which reduces reasoning tokens by $24.5\%$ across five mathematical datasets, within a $2.9\%$ drop in accuracy. [1]

## 1 INTRODUCTION

Large language models (LLMs) have made remarkable progress in reasoning abilities, achieving strong performance across domains such as mathematics, logic, and code synthesis (Cobbe et al., 2021; Chen et al., 2021). This leap is largely attributable to the development of Chain-of-Thought (CoT) reasoning pattern (Nye et al., 2021; Wei et al., 2022), which guides the model to break down complex problems into a series of intermediate steps. Recent breakthroughs such as OpenAI's o1 (OpenAI et al., 2024) and DeepSeek-R1 (DeepSeek-AI et al., 2025) have brought LLMs to the next paradigm, known as reasoning models (Ke et al., 2025; Zhang et al., 2025). Unlike traditional CoT reasoning, which follows a single linear thought process, reasoning models trained with Reinforcement Learning with Verifiable Rewards (RLVR) are believed to possess the ability to internally reflect on their reasoning steps, detect potential errors, and adaptively adjust the reasoning trajectories (Luo et al., 2025; Liu et al., 2025c; Yu et al., 2025; MiniMax et al., 2025; Li et al., 2025)

Within this paradigm, a consistent correlation is observed between the length of a model's generated response and its reasoning accuracy (Muennighoff et al., 2025). Models that generate more extensive CoTs tend to exhibit higher accuracy, suggesting that longer reasoning processes are more beneficial. One commonly observed pattern in these long reasoning rollouts is the presence of "reflections", where models elaborate or re-examine a solution after deriving a candidate answer. Intuitively, such reflections are assumed to be productive, much like human problem-solving, where self-reflections lead to correction or an "aha moment" (Chen et al., 2025b).

Despite their intuitive appeal, prior studies report mixed findings on the effects of reflective behaviors. Some emphasize the sophisticated internal mechanisms of reflection and their role in preventing reasoning collapse (Yang et al., 2025c), while others argue that self-reflection patterns are often superficial and do not improve outcomes (Liu et al., 2025c). Crucially, these studies provide limited

---

[1]The code and dataset of this work will be open-sourced upon acceptance.

**Problem statement:**
Let ABC be a triangle inscribed in circle ω. Let the tangents to ω at B and C intersect at point D, and let $\overline{AD}$ intersect ω at P. If AB = 5, BC = 9, and AC = 10, AP can be written as the form m/n , where m and n are relatively prime integers. Find m + n.

**Model Response:**

1: <think> Okay so I need to solve this geometry problem ...
...
368: Wait, that's a nice simplification! So $AP$ is 100/13. Therefore, $m = 100$ , $n = 13$ , which are coprime.
369: Wait, that seems straightforward. Let me check my steps again to ensure no mistakes.
...
519: Therefore, regardless of the method, AP is 100/13, so $m + n = 100 + 13 = 113$
520: But let me check once again with another approach to be thorough. Maybe using power of a point and some properties of triangle.
...
610: Therefore, $AP = DA − DP = 100/13$ Therefore, the answer is $100 + 13 = 113$. Thus, I can be confident that the correct answer is 113.
</think> To solve the problem ...

**Extractor Output:**

Candidate Position: 368
Candidate answer: 113

Candidate Position: 519
Candidate answer: 113

Candidate Position: 610
Candidate answer: 113

Figure 1: Illustration of a long CoT and the extraction result of candidate answers.

quantitative analysis on the reflective behavior of reasoning models, leaving unresolved whether reflections genuinely help models correct errors or merely confirm earlier conclusions.

To address this open question, we perform a systematic, large-scale quantitative study of reflection patterns in eight reasoning models across five mathematical benchmarks of varying difficulty. To extract these patterns, we design an LLM-based extractor that locates positions in the rollouts where candidate answers are produced. Since a rollout often contains multiple candidate answers, we define the portion of the rollout that follows the first candidate as reflections. This setup allows us to disentangle forward reasoning (steps leading to the first candidate) from reflective reasoning (subsequent steps) and to evaluate whether reflections genuinely contribute to error correction.

Our experiments quantitatively show that across various models and datasets, reflections are largely confirmatory and rarely corrective: once a candidate answer is proposed, subsequent reasoning steps seldom overturn it, instead mainly reiterating or justifying the initial answer. This finding challenges a wide belief that reflections are the primary mechanism for self-correction. It also raises two fundamental questions:

- *If reflections of reasoning models rarely change answers, why is their presence strongly correlated with accuracy?*
- *If reflections mostly confirm earlier conclusions, can we safely truncate them at inference time to reduce computation without significantly harming performance?*

To address these questions, we explore the role of reflections in both training and inference. On the training side, we conduct supervised fine-tuning (SFT) with datasets containing different amounts of reflective reasoning. Our results show that performance gains do not arise from teaching the model to self-correct after mistakes. Instead, **reflections in training data improve performance by increasing the likelihood that the model solves the problem correctly on the first try.** We hypothesize that rollouts with more reflections implicitly expose diverse reasoning paths toward the same problem, which enriches the training distribution and leads to better generalization on unseen problems. On the inference side, motivated by the observation that reflections are mostly confirmatory, we propose a simple yet effective strategy: early stopping of reflections when additional reasoning is unlikely to change the outcome. This method reduces token usage by $24.5\%$ with less than a $2.9\%$ drop in accuracy. Moreover, it allows a dynamic balance between token usage and performance by controlling the early stopping criteria. The contribution of this paper is three-fold:

- **Taxonomy of reflection behavior** (Section 2). We provide the first large-scale analysis of how reasoning models allocate tokens between forward reasoning and reflection, showing that reflections are mostly confirmatory rather than corrective, and the accuracy of the first try is the driving factor of improvement.
- **Training insights** (Section 3). We show that reflection-rich training data improve model accuracy by diversifying reasoning exposure and strengthening first-try correctness, not by enabling error correction.
- **Efficient inference technique** (Section 4). We propose an early-stopping method that reduces token consumption during inference and allows control over the balance between token usage and performance.

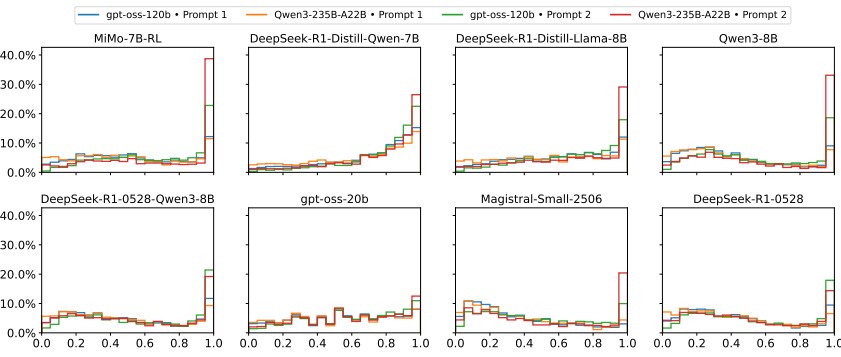

Figure 2: Distribution of first candidate answer positions across different LLMs and prompts. The x-axis denotes the relative position of the first candidate answer (line index divided by total lines), and the y-axis shows the proportion of rollouts in each bin.

## 2 ANALYZING REFLECTIONS IN REASONING MODELS

### 2.1 REFLECTION EXTRACTION

Conventional LLMs typically employ a single, linear generation process, concluding upon the initial derivation of a solution. Reasoning models, however, are capable of a more iterative and deliberative methodology (Ke et al., 2025). They can construct significantly more elaborated chain-of-thoughts (CoTs), not simply by extending the quantity of reasoning steps, but by generating and assessing multiple divergent reasoning trajectories and potential answers. This recursive process of refinement, frequently described as **"reflection"**, allows the model to compare alternatives, check intermediate claims, and potentially improve the final answer before committing.

While the capability of performing reflection often correlates with the reasoning capabilities of reasoning models, the internal mechanics of this process remain opaque. To unlock the full potential of these models and understand their decision-making, it is crucial to dissect their reflective patterns. We observe that within a long CoT, there can be multiple positions where the model has already derived a potential answer but opts to continue its reasoning before committing to a final output. Analyzing these critical points is fundamental to understanding the model's reflection process.

**Method** In this work, we define "reflection" as the contents occurring between two successive candidate answers in the reasoning process. To extract reflections, we introduce an LLM-based candidate answer extractor that parses long CoT outputs and identifies the positions of candidate answers, enabling a structured analysis of the model's reflective behavior. Specifically, a CoT can be represented as a sequence $C = \{s_1, s_2, ..., s_N\}$, where each $s_i$ is a reasoning step delimited by a line break. We employ an LLM (see Appendix A for prompts and example input), to extract plausible candidate answers, which can be formally expressed as follows:

$$\texttt{Extract}(C) = \{(i, a_i) \mid i \in [1, N] \wedge \texttt{IsCandidateAnswer}(s_i)\}, \tag{1}$$

where $\texttt{IsCandidateAnswer}(s_i)$ determines whether step $s_i$ contains a candidate answer, and $a_i$ denotes the extracted candidate answer. Note that the extraction process only requires understanding what quantity the question is asking, and whether a reasoning step derived it, without any requirement on the ability to actually solve the question. As a result, this process yields a structured set of candidate answers and their corresponding positions for subsequent analysis, shown in Figure 1.

**Setup** We apply our LLM-based extractor on the rollouts of five mathematical benchmarks: AIME2024 (MAA, 2024), AIME2025 (AIME, 2025), AMC (AMC12, 2025), Olympiad Bench (He et al., 2024), Math500 (Hendrycks et al., 2021). Among these benchmarks, Math500 is considered easier, with state-of-the-art models reaching more than $95\%$ accuracy, while AIME2024 and AIME2025 are considered harder, with model pass@1 performance ranging from $30\%$ to $80\%$. For benchmarks with fewer problems, such as AIME, we increase the number of rollouts per problem to ensure a robust and consistent evaluation (See Table 3 of Appendix B for details). In

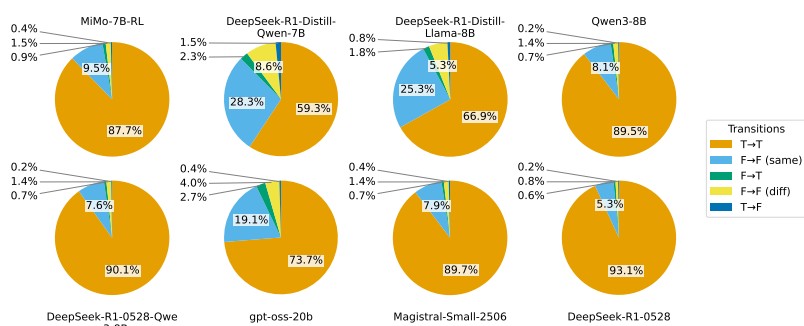

Figure 3: Reflections type statistics of long CoTs of different models. Long CoTs are collected on AIME2024 and AIME2025 (32 rollouts per question), AMC (4 rollouts per question), Olympiad Bench, and Math500 (1 rollout per question). Statistics are compiled for the union of all rollouts. More detailed breakdown of each dataset can be found in Figure 11 of Appendix D.

total, 3,427 rollouts are collected for each of the eight reasoning models evaluated. The studied models cover a wide spectrum of the reasoning model family, with sizes ranging from 7B to 685B, covering models trained with reinforcement learning (RL) (MiMo-7B-RL (Xiaomi et al., 2025), gpt-oss-20b (OpenAI, 2025), Magistral-Small-2506 (Mistral-AI et al., 2025), DeepSeek-R1-0528 (DeepSeek-AI et al., 2025)) and distillation (DeepSeek-R1-Distill-Qwen-7B (DeepSeek-AI et al., 2025), DeepSeek-R1-Distill-Llama-8B (DeepSeek-AI et al., 2025), Qwen3-8B (Yang et al., 2025a), DeepSeek-R1-0528-Qwen3-8B (DeepSeek-AI et al., 2025)).

**Robustness Analysis**  To evaluate the robustness and accuracy of our method, we first conduct a human evaluation. We randomly sample 100 rollouts and ask human participants to evaluate whether the candidate positions and candidate answers extracted by our extractor are reasonable. Across 100 rollouts with 426 extracted candidates in total, human participants labeled 94% of the candidate extraction as correct, which demonstrates the high reliability of the proposed extractor (See Appendix C for more details).

Further, we evaluate the sensitivity of our method by testing four extractor variants, constructed using two different LLMs (Qwen3-235B-A22B (Yang et al., 2025a) and gpt-oss-120b (OpenAI, 2025)) and two distinct extraction prompts (see Appendix A for both complete prompts). To analyze consistency across these configurations, we focus on the relative position of the first candidate answer, which serves as a stable and comparable reference point despite differences in the number and length of reflections across models. The histogram in Figure 2 shows that the distribution of the first candidate answer's relative position is consistent across different extractors for each model. This demonstrates the robustness and insensitivity of our extractor to the choice of LLMs or prompts, as it consistently captures reflection patterns across different models' outputs. We use gpt-oss-120b and prompt 1 in Appendix A for the extractor in the rest of the paper.

## 2.2 Reflection Analysis

**Reflection Types**  Using the LLM-based extractor, we identify and extract $n$ candidate answers for each CoT, here we denote them as $\{a_1, a_2, \ldots, a_n\}$ indexed by their appearing order. We then evaluate the correctness of each answer $a_i$ using a rule-based verifier [2], which returns `True` (T) if $a_i$ is correct and `False` (F) otherwise. For two consecutive candidate answers from the same CoT, $a_{i-1}$ and $a_i$, the **reflection type** is determined by whether the answer's correctness changes from the previous to the current attempt: (1) T $\to$ T: both $a_{i-1}$ and $a_i$ are correct; (2) F $\to$ F (same): both $a_{i-1}$ and $a_i$ are incorrect, and $a_{i-1} = a_i$; (3) F $\to$ T: $a_{i-1}$ is incorrect, and $a_i$ is correct; (4) F $\to$ F (diff): both $a_{i-1}$ and $a_i$ are incorrect, and $a_{i-1} \neq a_i$ ; (5) T $\to$ F: $a_{i-1}$ is correct, and $a_i$ is incorrect. Specifically, we define T $\to$ T and F $\to$ F (same) reflections as *confirmatory*, since the answer is not changed. And F $\to$ T reflections as *corrective*, since it changes an incorrect answer to correct.

---

[2]https://github.com/huggingface/Math-Verify

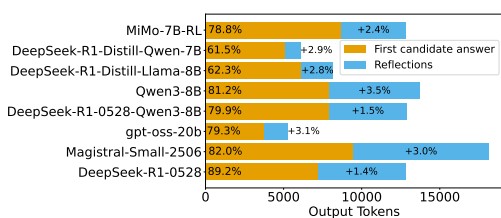

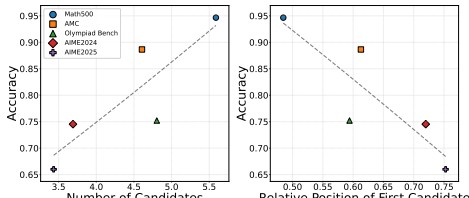

Figure 4: Breakdown of long CoTs: orange bars show the token count up to the first candidate answer, and blue bars show the token count in subsequent reflections. Numbers on bars indicate the accuracy of the first candidate answer, and the accuracy improvement brought by reflections.

Figure 5: Left: Average number of candidate answers per rollout across different datasets. Right: Relative position of the first candidate. Values are averaged over 8 models.

**Analysis on Reflection Types** The analysis of reflection types within long CoT, as depicted in Figure 3, reveals that over $90\%$ of the reflections are confirmatory, i.e., $T \to T$ and $F \to F$ (same), rather than corrective. This indicates that **most reflections reaffirm an existing answer instead of changing it**. This trend is universal across all tested models and datasets (see Figure 11 in Appendix D for a detailed breakdown). Crucially, the proportion of corrective reflections ($F \to T$) that actually improve performance is exceptionally small (mostly less than $2\%$).

**Impact of Reflection on Performance** To better understand the impact of reflections on performance and their token usage, we compare the accuracy of the first candidate answer with that of the final answer in the rollouts as shown in Figure 4. The accuracy of the first candidate answer is given on the orange bar segments, and the contribution of subsequent reflections is shown on the blue segments. Reported accuracies are averaged over AIME2024, AIME2025, AMC, Olympiad Bench, and Math500. We report this averaged accuracy unless mentioned otherwise throughout the paper. We observe that while reflections after the first candidate answer consume a large portion of the total tokens (ranging from $16.8\%$ to $47.8\%$), the resulting performance gain is limited (ranging from $1.4\%$ to $3.5\%$). This suggests that the final accuracy strongly correlates with the correctness of the first answer, highlighting its dominant influence. In other words, **the first try matters.** We provide a detailed breakdown in Tables 5 and 6 of Appendix G.

**Effect of Data Difficulty on Reflection Patterns** To analyze the reasoning models' reflection patterns across various mathematical datasets of different difficulties, we plot the average number of candidates and the average relative position of the initial candidate in Figure 5 (See Table 4 in Appendix E for detailed statistics). Our analysis reveals that on more challenging datasets, such as AIME2024 and AIME2025, the model allocates more tokens to forward reasoning, delaying the appearance of the first candidate. Conversely, on easier datasets such as Math500, the first candidate appears much earlier in the reasoning trajectory. This presents a counterintuitive pattern: **models perform more reflections on easier problems and fewer on difficult ones**, indicating that the reflection mechanism of reasoning models is not well aligned with task difficulty.

## 3 THE ROLE OF REFLECTION IN REASONING MODEL TRAINING

Analysis of reflection patterns in reasoning model rollouts reveals a counterintuitive phenomenon: the majority of reflections neither alter the candidate answer nor contribute meaningfully to performance improvement. This observation raises a critical question: why do reasoning models still achieve a substantial performance boost after being trained on long CoTs containing many confirmatory reflections? In this section, we study this by conducting supervised fine-tuning (SFT) on curated datasets with different reflection characteristics and comparing their performance.

### 3.1 TRAINING WITH VARYING AMOUNT OF REFLECTIONS

To investigate the role of reflection in reasoning, we begin by examining how the amount of reflections included in training data affects model performance.

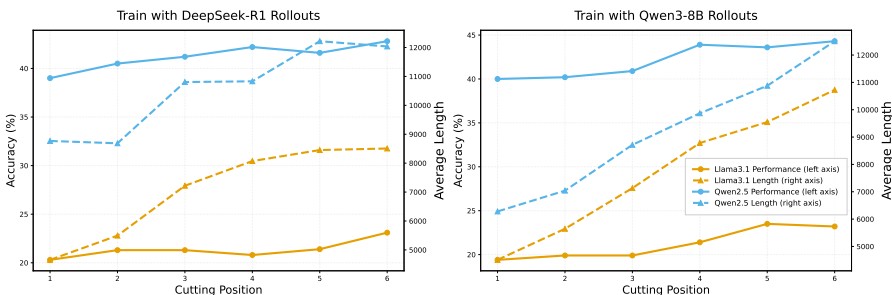

Figure 6: Comparison of performance and rollout length after SFT when training on rollouts cut at different positions. Qwen2.5-7B-Instruct and Llama3.1-8B-Instruct are trained using processed rollouts from DeepSeek-R1 and Qwen3-8B, respectively. Accuracies are averaged over five datasets.

**Training Data Construction** To study the effect of the number of reflections in long CoT data, we carefully manipulate the rollouts by truncating them after the occurrence of different candidate answers, thereby constructing a long CoT SFT dataset with a controllable number of reflections. Specifically, we use the MiroMind-M1-SFT dataset (Li et al., 2025), which contains mathematical problems curated from diverse sources, along with their corresponding DeepSeek-R1 rollouts. Additionally, we generate one rollout per problem using Qwen3-8B, providing an alternative rollout source for comparison. We then apply the candidate answer extractor to filter out rollouts that satisfy: (1) produce a correct final answer, and (2) the correct answer appears more than six times as a candidate. This filtering step ensures that each selected rollout includes sufficient reflections, allowing flexible truncation from the first to the sixth candidate answers to construct datasets with a varying number of reflections.

To create a rollout with exactly $i$ reflection steps, we truncate a filtered rollout at the $i$-th candidate answer, append the stop-thinking symbol `</think>`, then feed the truncated rollout into the reasoning model that used to generate it (e.g., DeepSeek-R1 or Qwen3-8B) to continue generation and produce the final answer. This yields a reasoning rollout with exactly $i$ reflection steps. See Appendix Figure 12 for an example. By continuing the truncated thinking, we ensure that the rollouts we used for training are still coherent, without abrupt stops. We filter out rollouts whose answers in the continued generation are different from the candidate answer from which we truncated. This step removes less than $0.5\%$ of rollouts, showing that once the model settles on a candidate answer, stopping the thinking process and prompting for a final response reliably yields that same answer.

Applying this procedure on the MiroMind-M1-SFT dataset, we curate six SFT datasets. The $i$-th dataset, termed the "cut-at-$i$" dataset, contains long CoTs truncated at the $i$-th reflection, resulting in exactly $i$ reflections per example. These datasets share the same set of problems, and all rollouts correctly solve the problem. The difference between them lies in the controlled number of reflections. Another variable we need to control is that, CoTs with more reflections typically contain more tokens. Therefore, for fair comparison, we downsample these datasets to ensure all six have the same number of training tokens. After this process, each generated dataset contains 28 million tokens, with the cut-at-1 dataset having 6,754 questions and the cut-at-6 dataset having 3,405 questions.

**Impact of Reflection Amount on SFT Performance** Given the curated dataset, we perform SFT on Llama3.1-8B-Instruct (Grattafiori et al., 2024), and Qwen2.5-7B-Instruct (Yang et al., 2024). We test on the combined set of AIME24, AIME25, Olympiad Bench, AMC, and Math500. The performance results and corresponding rollout lengths are presented in Figure 6. It shows that training on reflection-rich rollouts yields higher accuracy and longer generations across different datasets and model architectures. For SFT models, this suggests that, under a fixed token budget, constructing datasets with more reflection-rich rollouts is more effective than using the same budget to include more questions with shorter rollouts.

To better understand this improvement, we leverage the extractor and we split each rollout at the first candidate answer: the segment before it is denoted as "first candidate answer", and the segment after it as "reflections". We then compare their corresponding lengths and accuracies. Figure 7 (plotted in the same manner as Figure 4) shows results trained on Qwen3-8B rollouts, with similar DeepSeek-R1 results in Figure 13 of Appendix H.

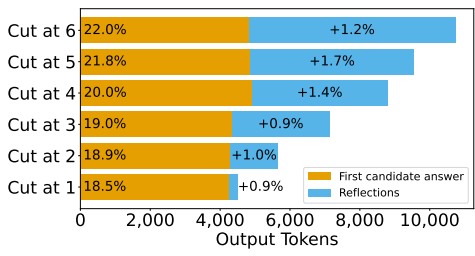 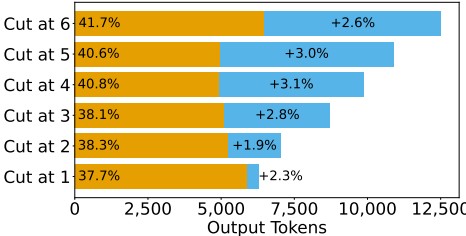

(a) Llama3.1-8B-Instruct          (b) Qwen2.5-7B-Instruct

Figure 7: Token usage and accuracy after SFT using Qwen3-8B rollouts. Before SFT, Llama3.1-8B-Instruct achieves 7.9% accuracy, and Qwen2.5-7B-Instruct achieves 35.3%.

Figure 7 demonstrates a clear trend that models trained on rollouts with more reflections achieve higher final performance. Averaging over Llama3.1-8B-Instruct and Qwen2.5-7B-Instruct, cut-at-6 outperforms cut-at-1 by $4.05\%$. This performance gain is mainly due to higher accuracy of the first candidate answers, which increase by an average of $3.75\%$ from cut-at-1 to cut-at-6, whereas the contribution from additional reflections is much smaller, averaging only $0.3\%$. Interestingly, while the accuracy of the first answer is increased, the token cost of generating the first answer remains consistent after trained with different cut-at-$i$ datasets. The tokens spent on reflections account for most of the difference, with an average of 5,636 reflection tokens increase per rollout from cut-at-1 to cut-at-6.

In conclusion, training models with more reflections leads to better performance and longer responses, which is expected. Surprisingly, from the breakdown of the source of improvements, we find that this gain does not come from reflections fixing incorrect answers, but from higher accuracy in the first candidate answer. One possible explanation is that richer reflections expose the model to diverse problem-solving approaches, improving generalization and boosting initial answer quality rather than simply correcting mistakes.

**Discussions** Our analysis in Figure 7 shows that SFT distillation enhances overall performance primarily by improving first-try correctness, especially when the SFT data includes more reflections, while the improvement brought by reflections is marginal. Our previous analysis in Figure 3 also shows that both RL-trained and SFT-distilled reasoning models show a similar pattern that reflections are mostly confirmatory and do not bring improvement. This raises the question of whether the RL training stage of reasoning models is also improving accuracy by making better first tries. To investigate this, we compare behaviors before and after RL of open-source reasoning models: MiroMind-M1-RL-7B (Li et al., 2025) and its initialization, MiroMind-M1-SFT-7B; MiroMind-M1-RL-32B and its initialization, DeepSeek-R1-Distill-Qwen-32B. As illustrated in Figure 8, we see that for both models, the performance gains after RL mainly come from first-answer accuracy improvement (+4.6% for 32B model, and +7.7% for 7B model), while gains attributable to reflections are marginal (+0.3% for 32B model, and +0.1% for 7B model). The experimental results indicate that during reinforcement learning, reasoning models primarily enhance their ability to produce a correct answer on the first attempt, rather than improving the quality of their subsequent reflections.

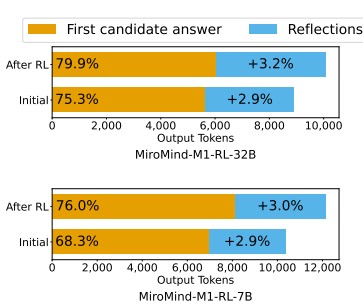

Figure 8: Changes of reasoning behavior after RL.

### 3.2 TRAINING WITH CORRECTIVE REFLECTION PATTERNS

In the previous section, we investigated how reflections affect model performance in SFT, and showed that more reflections in training rollout mainly improve first answer accuracy, with limited improvement on its corrective behavior. In this section, we test whether reflection ability can be improved by adding more corrective reflections (i.e., F → T) to the training dataset.

Table 1: Performance after SFT with different ratios of F → T reflections in the dataset.

| Ratio (%) | Llama3.1-8B-Instruct | | | Qwen2.5-7B-Instruct | | |
|---|---|---|---|---|---|---|
| | $p(\text{F} \rightarrow \text{T})$ | Accuracy (%) | Length | $p(\text{F} \rightarrow \text{T})$ | Accuracy (%) | Length |
| 100 | 0.053 | 26.6 | 10830 | 0.036 | 44.1 | 9655 |
| 75 | 0.050 | 25.6 | 11085 | 0.043 | 44.4 | 8775 |
| 50 | 0.058 | 27.3 | 10746 | 0.045 | 43.1 | 9943 |
| 25 | 0.059 | 26.2 | 11295 | 0.046 | 44.8 | 10094 |
| 0 | 0.050 | 26.9 | 11419 | 0.041 | 44.1 | 9363 |

**Dataset Construction**    We collect Qwen3-8B rollouts on math problems from the MiroMind-M1-SFT dataset. For each question, we sample one rollout containing at least one F → T reflection and one rollout consisting solely of T → T reflections. By filtering problems that have both type of rollouts (corrective and confirmatory), we kept 6K problems. Using these problems and their rollouts, we construct five datasets by varying the proportion of problems for which we select their corrective rollout to include in the dataset: 0%, 25%, 50%, 75%, and 100%. For the remaining problems in each dataset, we select their confirmatory rollouts to include in the dataset.

**Impact of Corrective Reflections on SFT Performance**    We perform SFT on Llama3.1-8B-Instruct and Qwen2.5-7B-Instruct using the constructed datasets, with results shown in Table 1. Models trained on datasets with varying proportions of corrective reflections show similar response lengths and accuracies. The performance difference between the best and worst models is just 1.7%, and the maximum difference in response length is only about 1K tokens. Moreover, their ability to flip an incorrect answer to a correct one, measured by $p(\text{F} \rightarrow \text{T})$, the probability that the next candidate is correct given that the current candidate is incorrect, shows no improvement. This indicates that training on rollouts containing corrective reflections is not more beneficial than training on rollouts with only confirmatory reflections. This echos with our earlier analysis, reasoning improvements are reflected mainly as higher first-answer accuracy, rather than increased $p(\text{F} \rightarrow \text{T})$.

## 4 EFFICIENT REASONING BY EARLY STOPPING

Our studies in Section 2 show that the reflections of reasoning models are primarily confirmatory, which suggests potential token efficiency gains by stopping once a few candidate answers are identified. In this section, we study the token-accuracy tradeoff under different early-stopping strategies. Specifically, we propose a question-aware adaptive early-stopping approach to improve token efficiency of the reasoning process.

**Candidate Answer Detector**    A straightforward way to reduce confirmatory reflection tokens is to monitor candidate answers and early-stop the reasoning process once a correct one is generated. To achieve this, we train a Qwen3-1.7B-based candidate answer detector (CAD) to detect for each sentence in the generation whether it contains the candidate answer. We construct CAD training data from annotated rollouts in the MiroMind-M1-SFT dataset. The sentences in each rollout, delimited by \n, are annotated by gpt-oss-120b and labeled 1 if they contain a candidate answer, or 0 otherwise. The CAD takes the corresponding question and one sentence in the rollout as input, and is trained to predict whether the sentence contains a candidate answer.

**Question-aware Reflection Controller**    While reflections are mostly confirmatory in reasoning rollouts, some mathematical problems may benefit more from reflections than others. To identify such problems and give them more reflection budget, we train a question-aware reflection controller (QRC) to predict for a problem whether we should stop at the first candidate, or allow more reflections before early-stopping. Specifically, we train a Qwen3-1.7B-based binary classifier that takes in only the problem statement, and output a binary label. The training data is collected from the annotated MiroMind-M1-SFT dataset, where a question is labeled 1 if its rollout contains F → T reflections, otherwise 0.

**Question-aware Adaptive Early-Stopping**    With CAD and QRC, we can reduce unnecessary reflections in the reasoning process through question-aware adaptive early-stopping. During inference, we first feed the question into the QRC to determine whether the reasoning process should terminate at the first candidate answer or do more reflections. Then we use CAD to monitor the appearance of

Table 2: Question-aware adaptive early-stopping improves token efficiency. Without QRC, reasoning terminates immediately after the first candidate answer is generated. The classification thresholds of QRC and CAD are set as 0.05 and 0.5, respectively.

| Dataset | Accuracy (%) | | | Length | | |
|---|---|---|---|---|---|---|
| | Qwen3-8B | +CAD | +CAD, +QRC | Qwen3-8B | +CAD | +CAD, +QRC |
| AIME2024 | 82.1 | 77.9 (-4.2) | 79.6 (-2.5) | 18,962 | 13,517 (-28.7%) | 14,869 (-21.6%) |
| AIME2025 | 70.8 | 65.0 (-5.8) | 65.8 (-5.0) | 22,998 | 17,664 (-23.2%) | 18,014 (-21.7%) |
| AMC | 93.0 | 90.0 (-3.0) | 89.4 (-3.6) | 13,279 | 8,432 (-36.5%) | 8,756 (-34.1%) |
| Math500 | 97.4 | 94.4 (-3.0) | 96.0 (-1.4) | 5,755 | 2,912 (-49.4%) | 3,593 (-37.6%) |
| Olympiad Bench | 80.2 | 76.9 (-3.3) | 78.4 (-1.8) | 14,633 | 10,479 (-28.4%) | 11,835 (-19.1%) |
| Average | 84.7 | 80.9 (-3.8) | 81.8 (-2.9) | 15,125 | 10,601 (-29.9%) | 11,414 (-24.5%) |

candidate answers during generation and terminate thinking accordingly. In practice, we terminate at the first candidate if QRC labeled 0, otherwise the third candidate. We apply this approach to Qwen3-8B reasoning model and report the performance on five mathematical datasets in Table 2.

Table 2 illustrates that CAD saves on average 29.9% tokens across five mathematical datasets, with a modest 3.8% drop. With QRC, the performance drop is improved to 2.9%, while still enjoying a 24.5% token reduction. By controlling the classification thresholds of CAD and the QRC, our method provides a handle to balance between performance and token usage. Figure 9 illustrates the trade-

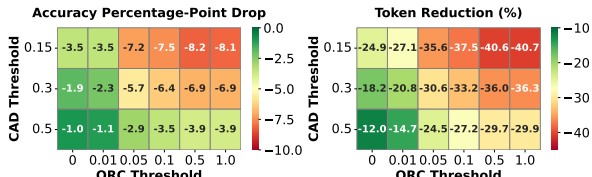

Figure 9: Accuracy drop and token reduction with varying classification thresholds of CAD and QRC.

off between token reduction and performance by adopting different threshold settings. On one extreme, a modest 1 percentage-point accuracy drop allows a 12.0% reduction in tokens; on the other extreme of the trade-off, an 8.12% accuracy drop corresponds to a 40.7% reduction in tokens.

## 5 RELATED WORKS

In the past year, scaling inference-time computation has emerged as a promising paradigm for improving the capabilities of large language models (LLMs) (Snell et al., 2024; Brown et al., 2025; Muennighoff et al., 2025). Building on this line of work, DeepSeek-R1 (DeepSeek-AI et al., 2025) demonstrates that inference-time scaling via reinforcement learning with verifiable rewards (RLVR) can unlock emergent reasoning abilities, yielding state-of-the-art results on challenging mathematical and coding benchmarks (Liu et al., 2025b; Ke et al., 2025; Xie et al., 2025; Jain et al., 2025).

Recently, a complementary body of research has examined why long chain-of-thought (CoT) reasoning is effective (Zhao et al., 2025; Chen et al., 2025a; Jiang et al., 2025). The dominant belief is that its success can be attributed to mechanisms such as recursive reflection, verification, and revision, which allow models to refine intermediate steps (Yang et al., 2025b; Wang et al., 2025). At the same time, excessively verbose traces are observed often to introduce redundancy, amplify hallucinations, and degrade the performance of reasoning models (Chen et al., 2025b; Zeng et al., 2025). In contrast, a separate line of literature argues that prolonged traces can strengthen reasoning ability and promote exploration of diverse solutions (Liu et al., 2025b;a). Despite these advances, the role of reflections within long CoTs remains underexplored. To address this gap, our work isolates and studies reflections within long CoTs across training and testing stages.

## 6 CONCLUSIONS

In this work, we systematically analyze the reflection pattern in long CoTs of reasoning models. We investigate their role in both the training and the inference phases. Through extensive experiments, we show that the reflections of reasoning models are mostly confirmatory, yet they are still helpful when included in training data. We also show that during inference time, confirmatory reflections consume a decent amount of tokens, while only introducing marginal improvements. To this end, we develop an efficient reasoning technique during inference to early stop excessive reflections while maintaining the performance. Together, these results provide a clearer understanding of the role of reflections and offer practical guidance for data design and inference efficiency.

ETHICS STATEMENT

We did not collect any new human-subject data or process personally identifiable information. No potential harms, such as bias amplification, toxic outputs, or dual-use risks, were observed in this work.

REPRODUCIBILITY STATEMENT

We are committed to reproducibility. All details of the data used and generated are documented in the paper, along with comprehensive descriptions of our experimental setup and hyperparameters. Upon publication, we will release the source code and data to the public to support transparent validation and further research.

USE OF LLMs

The use of LLM in this work is limited to rewriting sections to shorten them and save space for the page limit.

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

## A PROMPT FOR CANDIDATE EXTRACTION

**SYSTEM PROMPT 1**

```
SYSTEM PROMPT  Candidate Answer Extractor

Role: You read a math problem statement and a line-numbered model
    solution (thinking only).
Goal: For every line that presents a candidate answer to what the
    problem asks, output a record of (line_number,
    "candidate_answer_in_required_form"). Do not judge correctness
    or re-derive the solution.

1) Golden rule: lock the target first
Silently infer exactly what the original problem asks for:
- Target quantity: e.g., m+n, "remainder mod 1000", "sum of digits
    of N", "area", "number of solutions", etc.
- Required output form: e.g., integer, simplified fraction, decimal
    to k d.p., radical with squarefree radicand, gcd/coprime
    conditions, modulo residue, floor/ceil, units, etc.
- Trivial post-processing, if any (see ğ3).
Keep this target in working memory. All extraction converts to this
    target form.

2) What counts as a candidate answer
A line presents a candidate answer if it directly gives the target
    or uniquely determines it after only the trivial final steps in
    ğ3.

Include lines that:
- State the target explicitly (e.g., "Thus m+n=38", "Answer: 456").
- Give an equivalent numeric/expression that becomes the target
    after trivial conversion (e.g., a remainder before reduction,
    ab before radical normalization when the target is m+n, a
    fraction before simplification when the target is "lowest
    terms", a raw integer before taking "last 3 digits", etc.).
- Re-present the candidate (e.g., boxed, restated, "Therefore "),
    even if previously seen. Record every explicit presentation.

Exclude lines that:
- Only give intermediate facts not uniquely tied to the target
    (ranges/inequalities/bounds, generic identities, unspecialized
    parameters) unless the problem asks for those.
- Require nontrivial algebra, case analysis, multi-step geometry,
    or symbolic manipulation to reach the target (see ğ4).

3) Trivial Final Steps (you must perform these when applicable)
When a line yields a value that is one obvious step away from the
    target, perform the step and record the final target value:
- Simple arithmetic on explicit numerics/rationals
    (add/subtract/multiply/divide; reduce fractions to lowest
    terms).
- Modulo: compute N mod m; extract last k digits; compute parity.
- Digit ops: sum/product of digits; last digit.
- Floor/Ceil/Abs/Sign when directly evaluable on a numeric
    expression.
- Rounding exactly as requested (e.g., to 3 d.p.).
- Radicals normalization: rewrite ab with squarefree b, absorb
    perfect-square factors into a, ensure gcd(a,b)=1.
    - If the problem then asks for m+n, output that integer.
- Composite "reporting forms" common in contests:
    - If target is m+n from mn (squarefree n), or p+q from a
    reduced fraction p/q, compute and output the sum.
```

```
        - If target is "remainder", "units digit", "sum of
          coefficients", "sum of roots (given the polynomial)", etc., and
          the line gives the immediately-evaluable precursor, do the
          one-step conversion.
    - Unit conversion if it's a fixed scalar multiply/divide stated by
        the problem.
    Never cross into multi-step derivations. If it's more than a short,
        mechanical evaluation, do not include.

    4) Nontrivial (do not do)
    - No solving new equations, factoring beyond extracting perfect
        squares for radicals normalization, trig/geometry multi-steps,
        solving systems, case splits, or applying the quadratic formula
        unless it's already fully computed in the line.
    - No deducing implicit constraints unless the line states the value
        that pins the target after a trivial step.

    5) High-recall detection heuristics
    When scanning a line, look for any of the following cues. If
        present, attempt extraction.

    Textual cues:
    "so", "thus", "therefore", "hence", "we get", "equals", "is",
        "becomes", "gives", "yields", "implies", "it follows",
        "answer", "result", "final", "box/boxed".

    Math cues:
    - An equality or assignment (e.g., =, ,  if exact), explicit
        numerals, simplified forms, isolated expressions at the end of
        a derivation.
    - Named quantity matching the target (e.g., "remainder = 456", "sum
        of digits is 6").
    - Expressions that trivially map to target form (e.g., 1218 when
        target demands m+n).

    Repeat cues:
    If a line reasserts or updates a candidate, record it again with
        that line number.

    6) Per-line extraction algorithm (do this for each line
        independently)
    1. Collect candidates on this line:
        - Parse any explicit equalities/values/boxed content.
        - Note any expression that can be trivially converted to the
        target via ğ3.
    2. Resolve to target form:
        - Apply only ğ3 operations; otherwise stop.
        - If multiple possible candidates appear on the same line,
        record each separately.
    3. If successful, emit (line_number, "value_in_required_form").
    If no candidate survives ğ3, skip the line.

    7) Output format (STRICT)
    - Output a Python list of tuples only: [(line, "value"), (line,
        "value"), ...]
    - Keep tuples in the order of increasing line number; if multiple
        candidates on the same line, keep their left-to-right
        occurrence order.
    - "value" must be exactly what the problem asks for after trivial
        conversion (e.g., put "38", not "36*sqrt(2)" when the target is
        m+n).
    - The very last line of your reply must be only that list so eval()
        can parse it. No extra text.
```

```
8) Micro-examples (apply ğ3 automatically)
- Remainder: Line has N = 123456; target is N mod 1000 $\to$ record
    "456".
- Last 3 digits: Line has S = 7000456 $\to$ "456".
- Sum of digits: Line has N = 1002003 $\to$ "6".
- Reduced fraction: Line has 84/126 $\to$ "2/3".
- Radical m+n: Line has 1218 $\to$ normalize to 362 $\to$ m+n =
    36+2 = "38".
- Floor: Line has 7.99 and target is the integer part $\to$ "7".
Edge case principle: When in doubt, include if the target is
    uniquely determined by a single, trivial step.
```

## SYSTEM PROMPT 2

```
You are given a text block that contains the original problem
    statement, followed by a line-numbered "model solution".

Your job is **NOT** to judge correctness or solve the problem.
    Instead, read the solution **line by line** and record every
    line that presents a *candidate answer* to the problem. You
    need to fully understand what the problem asks for to notice
    the candidate answer. Only the thinking part of the model
    solution is provided for analysis.

Definitions
 *Candidate answer*  any explicit value or statement that (a)
    directly answers what the problem asks **or** (b) uniquely
    determines it with only a trivial final step (e.g. once you
    know N, taking "N mod 1000" is immediate).
 *Candidate answer* is not intermediate components (like individual
    addends when the question asks for their sum) unless the
    problem explicitly asks for each component.
 If the line gives an expression that still needs a trivial final
    computation to directly answer the question (for instance, a
    fraction whose numerator and denominator you must sum), carry
    out that simple arithmetic and record the result as your
    "candidate answer."
 There is likely multiple candidates answers in the model solution,
    and they are not necessarily the same as the model's final
    answer. You should not look for candidate answers by matching
    the model's final answer.

You can reason about the lines and decide whether they are
    candidate answers. For the final response, you should follow
    the format as below.

Final output format  strict
1. For each qualifying line output a two-element tuple:
        (line_number, "candidate_answer")
      `line_number` is an integer.
      `candidate_answer` is the exact answer text you extracted from
    that line (no boxing, no extra words) OR the answer that can be
    immediately implied from the line. Continuing the previous
    example, if the line indicates N=2016, the extracted candidate
    answer should be 16.
2. Collect the tuples in a Python list **in the order the lines
    appear**.
3. The **very last line** of your reply must be *only* that list,
    so that `eval()` can parse it, for example, [(12, "15"), (27,
    "3/4")]
```

```
4. Do **not** output anything after that list.
```

```
EXAMPLE INPUT

Analyze the following problem and its model solution.

----------------------------------------
Below is the problem statement **followed by** the line-numbered
    model solution:
----------------------------------------

Problem statement:
Find all prime numbers $p$ and positive integers $m$ such that
    $2p^2 + p + 9 = m^2.$
Model solution:
1: <think>
2: Okay, so I need to find all prime numbers p and positive
    integers m such that the equation 2pš + p + 9 = mš holds. Hmm,
    let's start by understanding the problem. I have to find primes
    p and positive integers m where this quadratic in p becomes a
    perfect square.
...
```

## B  BENCHMARK STATISTICS

Table 3: Statistics of Datasets and Rollouts

| Dataset | Problems | Rollouts per Problem | Total Rollouts |
|---|---|---|---|
| AIME 2024 | 30 | 32 | 960 |
| AIME 2025 | 30 | 32 | 960 |
| AMC | 83 | 4 | 332 |
| Olympiad Bench | 675 | 1 | 675 |
| MATH500 | 500 | 1 | 500 |
| Total | 1318 | - | 3427 |

## C  HUMAN EVALUATION

To validate our extraction method, we conduct a human annotation study with four participants. We task the annotators with labeling the correctness of each candidate answer in the rollout detected by our model. Each candidate is presented with the sentence that contains a candidate answer, alongside the corresponding problem statement of the rollout and the ground-truth solution.

The evaluation centers on two key questions, as illustrated in our user interface (Figure 10):

- **Q1:** This question tests whether the model correctly identifies a sentence containing a candidate answer. For example, a sentence such as "So the answer is 5?" qualifies as a valid candidate, whereas "Let's try to solve this" does not.

- **Q2:** This question assesses whether the extracted candidate answer is an answer to the question, regardless of correctness. For example, if the question is asking some quantity that "can be represented as m/n, what is m+n", then an extraction of "5" is valid in form, while "m=2, n=3" is invalid, as it does not allow the math verifier to robustly evaluate its correctness.

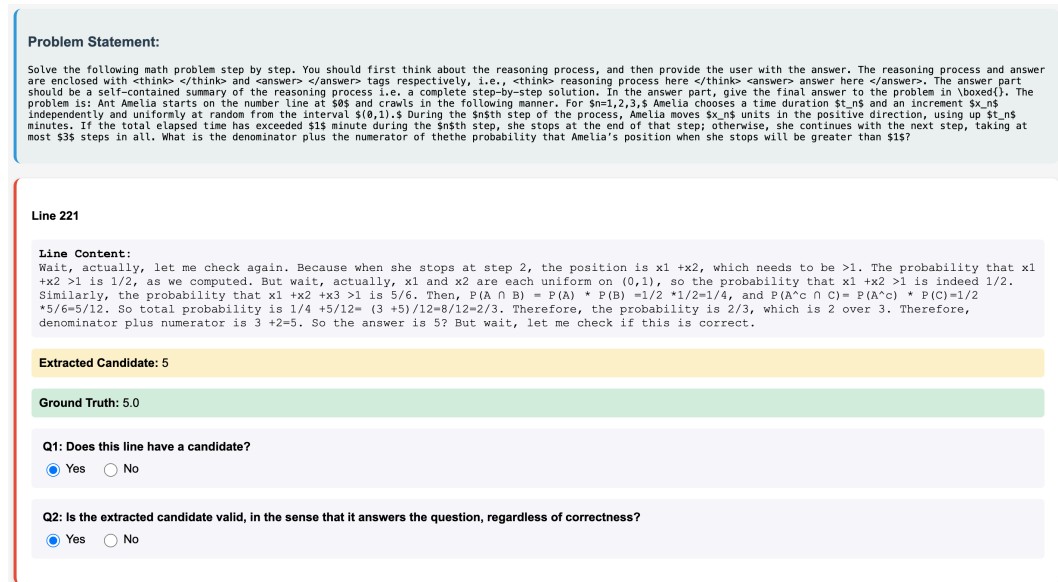

Figure 10: The user interface for evaluating LLM extraction correctness for human participants.

Based on human verification of 100 randomly sampled rollouts (comprising 426 candidates), our model demonstrates high performance. It achieves 94.1% accuracy in identifying the correct position (Q1) and 94.0% accuracy in adhering to the target format (Q2).

# D    BREAKDOWN OF REFLECTIONS

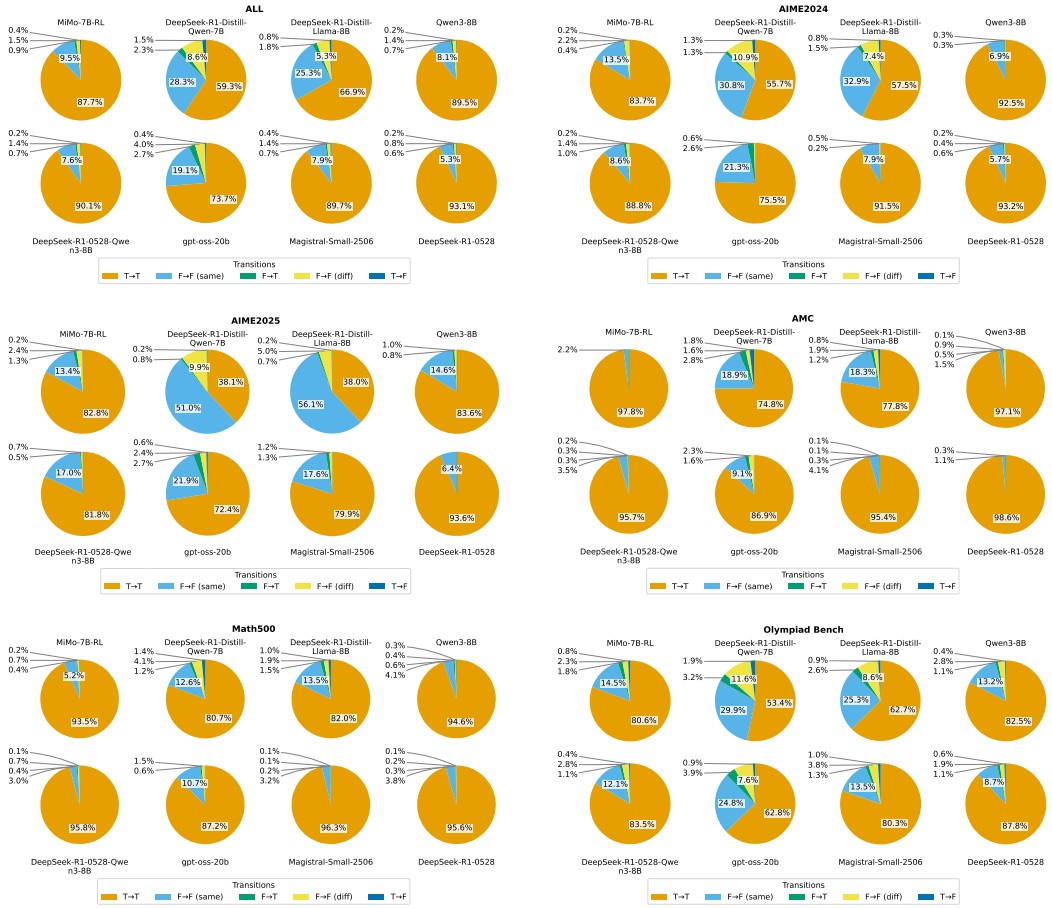

Figure 11: Reflection statistics of long CoTs of different models. Long CoTs are collected on AIME24 and AIME25 (32 rollouts per question), AMC (4 rollouts per question), Olympiad Bench, and Math500 (1 rollout per question).

# E    REFLECTION STATISTICS

Table 4: Number of candidate answers in rollouts across different reasoning models and datasets.

| Model | AIME2024 | AIME2025 | Math500 | Olympiad Bench | AMC |
|---|---|---|---|---|---|
| MiMo-7B-RL | 3.64 | 3.52 | 5.79 | 4.69 | 4.64 |
| DeepSeek-R1-Distill-Qwen-7B | 3.50 | 3.12 | 2.47 | 3.76 | 2.96 |
| DeepSeek-R1-Distill-Llama-8B | 3.84 | 3.87 | 4.52 | 4.59 | 4.38 |
| Qwen3-8B | 4.27 | 3.72 | 6.94 | 6.10 | 5.77 |
| DeepSeek-R1-0528-Qwen3-8B | 3.55 | 3.17 | 6.32 | 4.74 | 4.57 |
| gpt-oss-20B | 3.22 | 3.12 | 3.01 | 3.48 | 3.31 |
| Magistral-Small-2506 | 3.85 | 4.00 | 9.32 | 6.23 | 6.19 |
| DeepSeek-R1-0528 | 3.65 | 2.96 | 6.34 | 4.85 | 5.03 |

## F  EXAMPLE OF SFT DATA CURATION

Figure 12: An illustration of the SFT data curation process in Section 3.1.

## G  ROLLOUT ANALYSIS

Table 5: Rollout analysis of 8 models on 5 datasets. "First" and "Final" indicates the accuracy of the first appearing candidate and the final answer of the rollout.

| Model | AIME2024 First | AIME2024 Final | AIME2025 First | AIME2025 Final | AMC First | AMC Final | Olympiad Bench First | Olympiad Bench Final | Math500 First | Math500 Final | Average First | Average Final |
|---|---|---|---|---|---|---|---|---|---|---|---|---|
| MiMo-7B-RL | 72.5 | 72.5 | 65.0 | 67.9 | 89.2 | 89.2 | 74.3 | 80.3 | 93.1 | 96.2 | 78.8 | 81.2 |
| DeepSeek-R1-Distill-Qwen-7B | 53.3 | 52.1 | 33.8 | 35.4 | 76.4 | 79.2 | 59.0 | 64.3 | 85.2 | 90.8 | 61.5 | 64.4 |
| DeepSeek-R1-Distill-Llama-8B | 52.5 | 53.3 | 33.1 | 34.3 | 81.0 | 81.6 | 60.7 | 67.4 | 84.4 | 89.0 | 62.3 | 65.1 |
| Qwen3-8B | 81.7 | 82.1 | 67.9 | 70.8 | 91.6 | 92.8 | 72.7 | 80.3 | 92.2 | 97.4 | 81.2 | 84.7 |
| DeepSeek-R1-0528-Qwen3-8B | 76.2 | 77.9 | 66.2 | 68.8 | 92.8 | 90.7 | 71.5 | 74.9 | 92.3 | 94.9 | 79.8 | 81.4 |
| gpt-oss-20b | 76.4 | 78.5 | 74.5 | 77.0 | 87.0 | 89.4 | 67.7 | 73.5 | 90.9 | 93.9 | 79.3 | 82.4 |
| Magistral-Small-2506 | 83.8 | 84.2 | 70.4 | 72.9 | 93.2 | 93.5 | 68.9 | 77.5 | 94.0 | 97.0 | 82.0 | 85.0 |
| DeepSeek-R1-0528 | 89.6 | 90.0 | 85.8 | 86.2 | 96.4 | 95.8 | 78.4 | 82.6 | 95.2 | 98.4 | 89.1 | 90.6 |

Table 6: Rollout analysis of 8 models on 5 datasets. "FC" stands for first candidate, indicating the token usage of getting the first candidate. "Refl." stands for reflection, indicating the token usage of reflection after first candidate.

| Model | AIME2024 FC | AIME2024 Refl. | AIME2025 FC | AIME2025 Refl. | AMC FC | AMC Refl. | Olympiad Bench FC | Olympiad Bench Refl. | Math500 FC | Math500 Refl. | Average FC | Average Refl. |
|---|---|---|---|---|---|---|---|---|---|---|---|---|
| MiMo-7B-RL | 11,728 | 2,320 | 13,437 | 1,910 | 7,186 | 2,287 | 7,964 | 2,815 | 3,146 | 1,868 | 8,692 | 2,240 |
| DeepSeek-R1-Distill-Qwen-7B | 7,259 | 1,116 | 9,085 | 991 | 3,712 | 752 | 3,912 | 1,167 | 1,339 | 444 | 5,061 | 894 |
| DeepSeek-R1-Distill-Llama-8B | 9,404 | 1,256 | 9,862 | 1,237 | 4,434 | 1,246 | 5,126 | 1,533 | 1,920 | 1,024 | 6,149 | 1,259 |
| Qwen3-8B | 10,723 | 3,676 | 12,254 | 3,705 | 6,597 | 3,472 | 7,492 | 3,326 | 2,522 | 2,345 | 7,918 | 3,305 |
| DeepSeek-R1-0528-Qwen3-8B | 10,907 | 3,090 | 11,897 | 2,568 | 6,631 | 2,597 | 7,507 | 2,602 | 2,558 | 2,309 | 7,900 | 2,633 |
| gpt-oss-20b | 5,814 | 923 | 5,861 | 1,242 | 2,560 | 738 | 3,539 | 1,734 | 1,039 | 386 | 3,763 | 1,005 |
| Magistral-Small-2506 | 12,539 | 6,424 | 14,272 | 6,410 | 8,312 | 6,830 | 8,654 | 6,785 | 3,547 | 5,936 | 9,465 | 6,477 |
| DeepSeek-R1-0528 | 9,232 | 3,450 | 11,823 | 3,558 | 6,065 | 3,267 | 6,586 | 3,100 | 2,241 | 2,194 | 7,189 | 3,114 |

## H    TOKEN USAGE AND ACCURACY

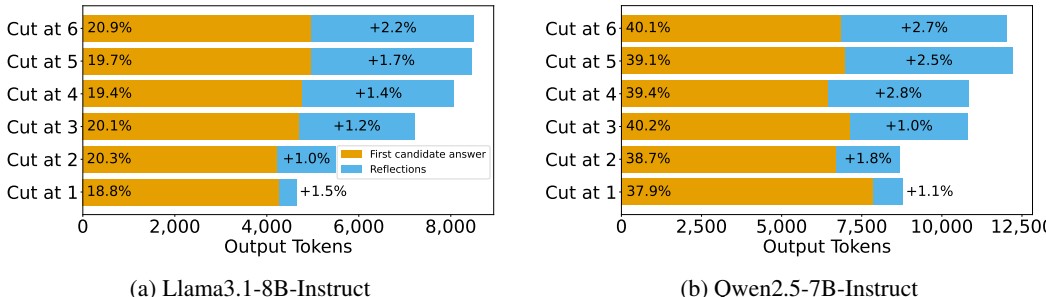

(a) Llama3.1-8B-Instruct  (b) Qwen2.5-7B-Instruct

Figure 13: Token usage and accuracy after SFT using Deepseek-R1 rollouts, separated into two parts: the first candidate answers (before the first candidate) and the reflections (after the first candidate). In the figure, orange bars show tokens used before the first candidate answer, blue bars show tokens used for reflections, and performance is marked on each segment. Cut at i rows correspond to models trained with R1 rollouts truncated at different positions. Training data are controlled so that they have same number of training tokens. We can see that cut-at-6 is on average 3.8% better than cut-at-1 (averaging over two models), while the improvement in first answer accuracy contributes 2.65% improvement, and reflection contributes 1.15% improvement.

## I    COMPARING DIVERSITY BETWEEN ROLLOUTS

**Diversity Prompt**

```
Please compare the two response and judge whether Response 2
    contains any new mathematical concept, or is simply redoing the
    previous calculations. Note, "new mathematical concept" can be
    any new information, imagine this is educational material to
    someone not familiar with math. Will Response 2 provide more
    information to them instead of simply repeating some previous
    calculation already in Response 1? In the last line of your
    response, please indicate "<YES>" or "<NO>".
```

To assess whether datasets truncated at later positions include extra useful information, we compare two responses to the same problem: one from dataset cut-at-{i} and the other from dataset cut-at-{i+1}, where the latter contains one more piece of reflection. Using an LLM (gpt-oss-120b), we evaluate whether this added text introduces any new mathematical concepts or problem-solving approaches. For all problems, we then compute the percentage for which cut-at-{i+1} contributes additional concepts/approaches beyond cut-at-{i}. The statistics are shown in Table 7

Table 7: Diversity analysis of longer rollouts. Percentage stands for the percentage of problems that the longer rollouts see additional problem-solving approaches.

|  | Percentage (%) |
| --- | --- |
| From cut-at-1 to cut-at-2 | 22.6 |
| From cut-at-2 to cut-at-3 | 16.8 |
| From cut-at-3 to cut-at-4 | 15.6 |
| From cut-at-4 to cut-at-5 | 16.4 |
| From cut-at-5 to cut-at-6 | 15.2 |

We provide two cases showing when the rollout with more reflections provides additional problem-solving approaches (Figure 14) and when the rollout with more reflections repeats previous calculations (Figure 15).

**Question.** Even or odd sum of all natural numbers from 1 to 17?

**Response in cut-at-1**

<think> Okay, so I need to figure out whether the sum of all natural numbers from 1 to 17 is even or odd. Hmm, let's start by recalling how to calculate the sum of numbers from 1 to a certain number. Let me remember... Oh yeah, it's the formula for the sum of an arithmetic series. The formula is n(n + 1)/2, where n is the last number in the sequence. So in this case, n is 17. Let me write that down:

Sum = 17 * (17 + 1) / 2

Calculating that, first I need to compute 17 + 1, which is 18. Then multiply by 17: 17 * 18. Hmm, what's 17 times 18? Let me do that step by step. 17 * 10 is 170, 17 * 8 is 136, so adding them together: 170 + 136 = 306. Then divide by 2: 306 / 2 = 153. So the total sum is 153. Now, I need to check if 153 is even or odd.

An even number is divisible by 2 without a remainder, and an odd number isn't. So, let me divide 153 by 2. 2 goes into 153 how many times? Well, 2*70 is 140, subtract that from 153, you get 13. Then 2*6 is 12, so that's 76 with a remainder of 1. So 153 divided by 2 is 76 with a remainder of 1, which means it's an odd number. Therefore, the sum is odd. ... Actually, that's correct, no further checking needed, I believe I have got the answer. Let's stop thinking and give a concrete answer based on the thoughts.

**Response in cut-at-2**

<think> Okay, so I need to figure out whether the sum of all natural numbers from 1 to 17 is even or odd. Hmm, let's start by recalling how to calculate the sum of numbers from 1 to a certain number. I think there's a formula for that. Let me remember... Oh yeah, it's the formula for the sum of an arithmetic series. The formula is n(n + 1)/2, where n is the last number in the sequence. So in this case, n is 17. Let me write that down:

Sum = 17 * (17 + 1) / 2

Calculating that, first I need to compute 17 + 1, which is 18. Then multiply by 17: 17 * 18. Hmm, what's 17 times 18? Let me do that step by step. 17 * 10 is 170, 17 * 8 is 136, so adding them together: 170 + 136 = 306. Then divide by 2: 306 / 2 = 153. So the total sum is 153. Now, I need to check if 153 is even or odd.

An even number is divisible by 2 without a remainder, and an odd number isn't. So, let me divide 153 by 2. 2 goes into 153 how many times? Well, 2*70 is 140, subtract that from 153, you get 13. Then 2*6 is 12, so that's 76 with a remainder of 1. So 153 divided by 2 is 76 with a remainder of 1, which means it's an odd number. Therefore, the sum is odd.

Wait, but maybe there's another way to figure this out without calculating the entire sum? Let me think. Since we're dealing with parity (even or odd), maybe we can find the number of odd numbers in the sequence from 1 to 17 and see if their count is odd or even. Because adding an even number of odd numbers would result in an even sum, and adding an odd number of odd numbers would result in an odd sum. But then again, even numbers don't affect the parity. So let's see.

From 1 to 17, how many odd numbers are there? Well, the numbers alternate between odd and even. Starting with 1 (odd), then 2 (even), 3 (odd), etc. Since 17 is odd, there are (17 + 1)/2 = 9 odd numbers? Wait, let me check. If you have numbers from 1 to n, the number of odd numbers is ceil(n/2). For example, if n is even, like 16, then there are 8 odd numbers. If n is odd, like 17, then there are (17 + 1)/2 = 9 odd numbers. Let me count them manually to confirm. Numbers from 1 to 17:

1 (odd), 2 (even), 3 (odd), 4 (even), 5 (odd), 6 (even), 7 (odd), 8 (even), 9 (odd), 10 (even), 11 (odd), 12 (even), 13 (odd), 14 (even), 15 (odd), 16 (even), 17 (odd). Let me count how many odds: 1,3,5,7,9,11,13,15,17. That's 9 numbers. So 9 odd numbers.

Now, each odd number contributes 1 to the parity of the sum. So adding 9 odd numbers together. Since 9 is odd, the sum of 9 odd numbers is odd. Because adding two odd numbers gives even, adding another odd number makes it odd again. So odd + odd = even, even + odd = odd. So for 9 odds, since 9 is odd, the total sum would be odd. Therefore, the sum from 1 to 17 is odd.

... Actually, that's correct, no further checking needed, I believe I have got the answer. Let's stop thinking and give a concrete answer based on the thoughts.

Figure 14: One case where the additional reflection provides another problem-solving approach: To know the parity of the sum, one only needs to count the number of odd numbers.

---

**Question.** What is the number? If I subtract a number from 12, I get 7. Calculate!

**Response in cut-at-1**

<think> Okay, let's see. The problem says, "If I subtract a number from 12, I get 7. Calculate!" So I need to find the number that when subtracted from 12 gives 7. Hmm, let me break this down.

First, let me translate the words into a mathematical equation. The problem states that subtracting a number from 12 results in 7. Let me denote the unknown number as x. So, the equation would be 12 minus x equals 7. That is, 12 - x = 7.

Now, I need to solve for x. To do that, I can rearrange the equation. Let me think. If I subtract x from 12, the result is 7. So, to isolate x, I can subtract 7 from both sides? Wait, no. Let me recall how to solve linear equations.

Starting with 12 - x = 7. If I want to solve for x, I can add x to both sides to get rid of the negative sign. So, adding x to both sides gives 12 = 7 + x. Then, to get x by itself, I subtract 7 from both sides. That would be 12 - 7 = x. Calculating that, 12 minus 7 is 5. So x equals 5.

... Actually, that's correct, no further checking needed, I believe I have got the answer. Let's stop thinking and give a concrete answer based on the thoughts.

**Response in cut-at-2**

<think> Okay, let's see. The problem says, "If I subtract a number from 12, I get 7. Calculate!" So I need to find the number that when subtracted from 12 gives 7. Hmm, let me break this down.

First, let me translate the words into a mathematical equation. The problem states that subtracting a number from 12 results in 7. Let me denote the unknown number as x. So, the equation would be 12 - x = 7.

Now, I need to solve for x. To do that, I can rearrange the equation. Let me think. If I subtract x from 12, the result is 7. So, to isolate x, I can subtract 7 from both sides? Wait, no. Let me recall how to solve linear equations.

Starting with 12 - x = 7. If I want to solve for x, I can add x to both sides to get rid of the negative sign. So, adding x to both sides gives 12 = 7 + x. Then, to get x by itself, I subtract 7 from both sides. That would be 12 - 7 = x. Calculating that, 12 minus 7 is 5. So x equals 5.

Wait, let me check that again. If I subtract 5 from 12, does that give 7? Let me verify. 12 minus 5 is indeed 7. Yeah, that works. So the number must be 5.

... Actually, that's correct, no further checking needed, I believe I have got the answer. Let's stop thinking and give a concrete answer based on the thoughts.

Figure 15: One case where the additional reflection is simple repetition.

## J TESTS ON NON-MATHEMATICAL DOMAINS

In addition to mathematical problems, in this section, we analyze the reasoning rollouts of Qwen3-8B, Qwen3-14B, and DeepSeek-R1-Distill-Llama-8B on three non-mathematical domains from MMLU: Formal Logic, High School Statistics, and Economics. Since MMLU are multiple-choice questions, we use the following prompt to extract the candidate answer positions.

**System Prompt for MMLU**

```
You are given a text block that contains the original problem
    statement and a list of four choices to choose from to answer
    the problem, followed by a line-numbered model solution.

Your job is **NOT** to judge correctness or solve the problem.
    Instead, read the solution **line by line** and record every
    line where the model has implicitly or explicitly derived a
    candidate answer to the problem (it may not be correct, as long
    as it is an answer to the problem it counts). You need to fully
    understand what the problem asks for to notice the candidate
    answer. Only the thinking part of the model solution is
    provided for analysis.

Definitions
 *Candidate answer*  any explicit value or statement that (a)
    directly answers what the problem asks **or** (b) uniquely
    determines it with only a trivial final step (e.g. if the
    problem asks N mod 1000, then once you know N, taking N mod
    1000 is immediate).
```

```
 *Candidate answer* is not intermediate components (like individual
    addends when the question asks for their sum) unless the
    problem explicitly asks for each component.
 There is likely multiple candidates answers in the model solution,
    and they are not necessarily the same as the model's final
    answer. You should not look for candidate answers by matching
    the model's final answer.

 You can reason about the lines and decide whether they are
    candidate answers. For the final response, you should follow
    the format as below.

 Final output format  strict
 1. For each qualifying line output a two-element tuple:
        (line_number, choice)
      `line_number` is an integer.
      `choice` is one of A, B, C or D.
 2. Collect the tuples in a Python list **in the order the lines
    appear**.
 3. The **very last line** of your reply must be *only* that list,
    so that `eval()` can parse it, for example, [(12, "A"), (27,
    "C")]
 4. Do **not** output anything after that list.
```

We report results of Qwen3-8B, Qweb3-14B, and DeepSeek-R1-Distill-Llama-8B on three MMLU tasks in Table 8, 9, and 10. We can see that on these non-mathematical tasks, the models exhibit a similar pattern as in mathematical tasks, where the first answer accuracy closely matches the final answer accuracy, and the relative position of the first answer is early.

Table 8: Performance of Qwen3-8B on three MMLU tasks.

| Qwen3-8B | Formal Logic | High School Statistics | Economics |
|---|---|---|---|
| First answer accuracy | 94.4 | 91.7 | 79.8 |
| Final answer accuracy | 96.8 | 94.4 | 80.7 |
| First answer relative position | 0.47 | 0.51 | 0.54 |

Table 9: Performance of Qwen3-14B on three MMLU tasks.

| Qwen3-14B | Formal Logic | High School Statistics | Economics |
|---|---|---|---|
| First answer accuracy | 96.8 | 93.1 | 78.9 |
| Final answer accuracy | 97.6 | 94.4 | 80.7 |
| First answer relative position | 0.51 | 0.56 | 0.59 |

Table 10: Performance of DeepSeek-R1-Distill-Llama-8B on three MMLU tasks.

| DeepSeek-R1-Distill-Llama-8B | Formal Logic | High School Statistics | Economics |
|---|---|---|---|
| First answer accuracy | 64.3 | 77.8 | 45.6 |
| Final answer accuracy | 66.7 | 79.2 | 48.2 |
| First answer relative position | 0.66 | 0.69 | 0.74 |

## K  DISCUSSION ON REINFORCEMENT LEARNING STAGE

In Section 3, we studied the role of reflections in supervised-fine-tuning stage of the model training, and briefly discussed the implications on RL stage. In this section, we further discuss how our findings may apply to the reinforcement learning stage.

We first conducted an RL training experiment, training a model initialized from Qwen2.5-Math-7B with the DeepScaleR dataset and GRPO algorithm. We collect rollouts on the five mathematical

datasets studied for every 5 steps of RL training and analyze their first answer accuracy and final answer accuracy. As shown in Figure 16 the first-answer-accuracy closely matches the final answer accuracy as training progresses. The echos with our findings that performance of a model is mainly decided by its first answer accuracy, and models performance boost from RL also mainly originates from the boost in first answer accuracy.

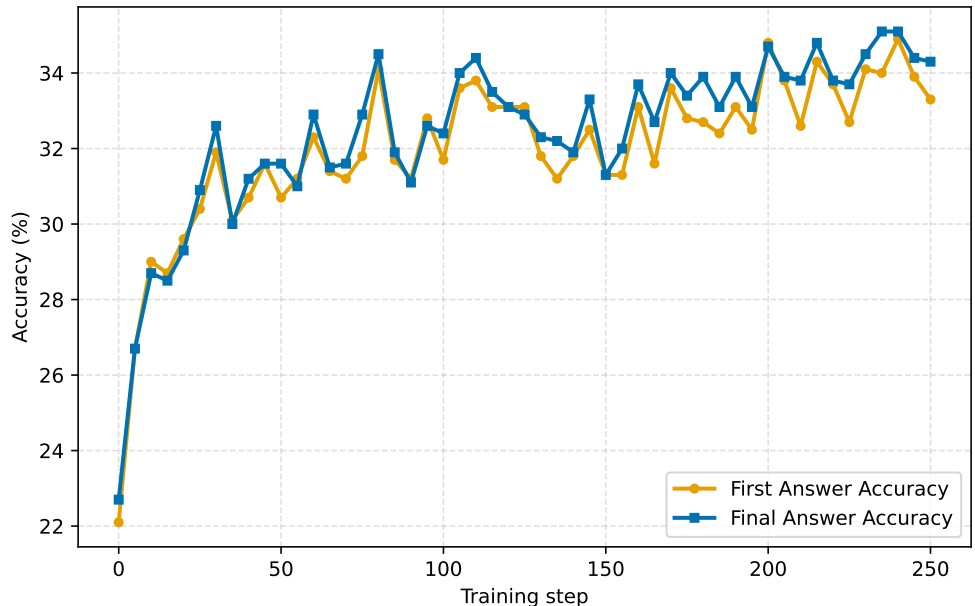

Figure 16: Change of first answer accuracy and final accuracy as RL training progresses.

## L    TRANSFERABILITY OF CAD AND QRC

In this section, we study the transferability of the proposed candidate answer detector (CAD) and question-aware reflection controller (QRC) on other models and datasets. To do that, we use the same CAD and QRC trained on the LLM-annotated Qwen3-8B rollouts on questions in M1-SFT dataset, and apply to two models, Qwen3-14B and DeepSeek-R1-Distill-Llama-8B, on the five mathematical datasets studied in the paper (Math500, AMC, Olympiad Bench, AIME2024, and AIME2025), as well as three recent mathematical datasets at the time of writting: CMIMC, HMMT, and BRUMO.

We report the results in Figure 17. We can see that the CAD and QRC can transfer well when applied to Qwen3-14B on different datasets. Suggesting its robustness to model and problem formatting. Moreover, we also see that in some threshold configurations of CAD and QRC (e.g. 0.5 and 0), we achieve a 1.9% performance increase while reducing 13.5% of the tokens by early stopping. Suggesting that excessive reflections may even have negative effects on the models reasoning performance. Additionally, we show that CAD and QRC is not sensitive to problem difficulty by showing their performance on AIME2025, which is considered the hardest among the five studied datasets. Comparing Figure 17c with 17e and 17d with 17f, we can see that on AIME2025, CAD and QRC do not introduce more performance drop than average, suggesting their robustness to problem difficulties.

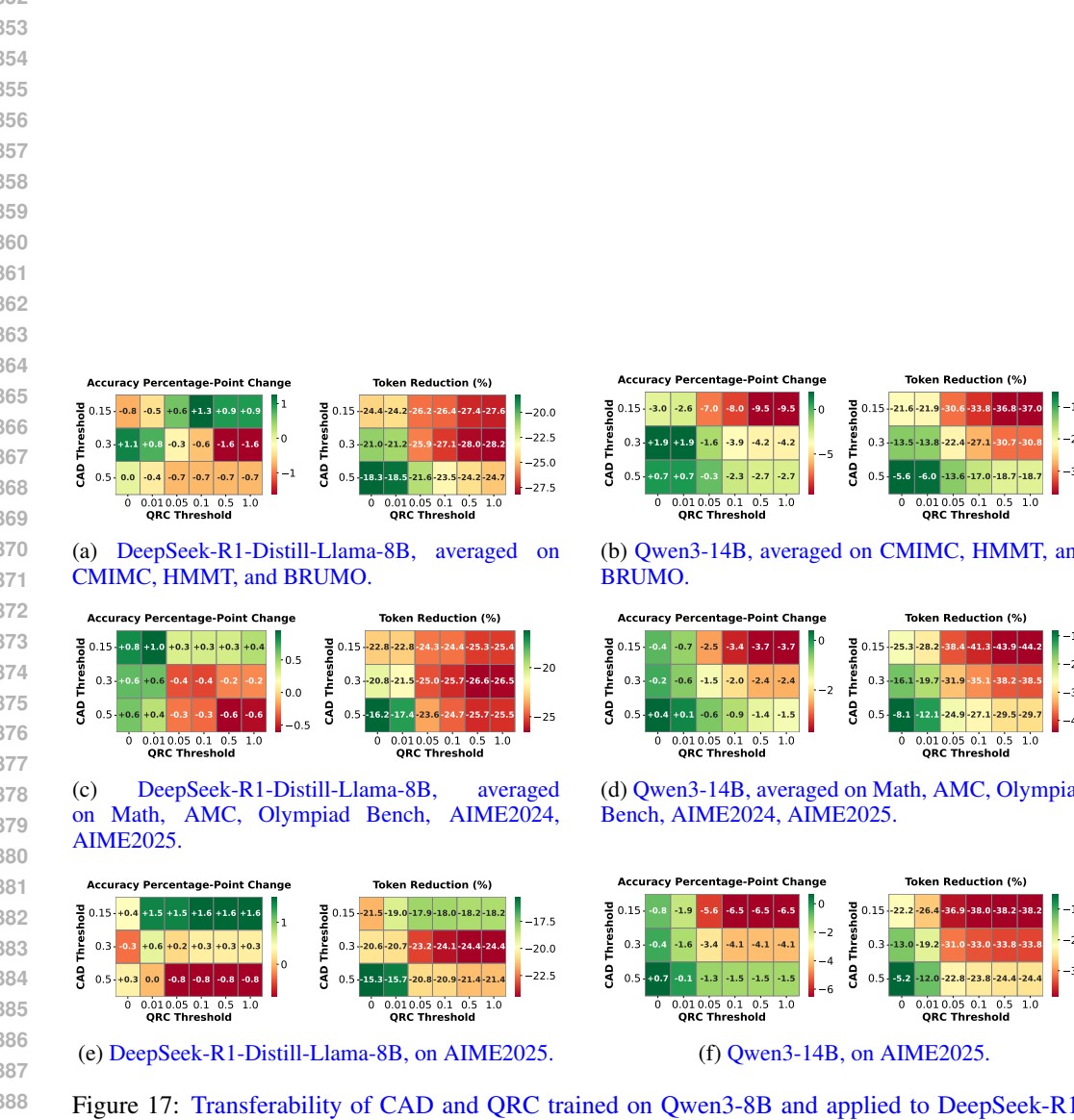

(a) DeepSeek-R1-Distill-Llama-8B, averaged on CMIMC, HMMT, and BRUMO.

(b) Qwen3-14B, averaged on CMIMC, HMMT, and BRUMO.

(c) DeepSeek-R1-Distill-Llama-8B, averaged on Math, AMC, Olympiad Bench, AIME2024, AIME2025.

(d) Qwen3-14B, averaged on Math, AMC, Olympiad Bench, AIME2024, AIME2025.

(e) DeepSeek-R1-Distill-Llama-8B, on AIME2025.

(f) Qwen3-14B, on AIME2025.

Figure 17: Transferability of CAD and QRC trained on Qwen3-8B and applied to DeepSeek-R1-Distill-Llama-8B and Qwen3-14B.

