# OpenReview forum: "First Try Matters: Revisiting the Role of Reflection in Reasoning Models"
_ICLR.cc/2026/Conference — Submitted to ICLR 2026_

### Official Review · Reviewer_hGc3 · 2025-10-27

**Soundness:** 2
**Presentation:** 3
**Contribution:** 3
**Rating:** 6
**Confidence:** 5

**Summary:**

This paper presents a systematic and quantitative investigation into the reflection behavior of reasoning models. By disentangling forward and reflective reasoning through an LLM-based extractor, the authors provide empirical insights into how reflections impact both training and inference. The work is novel and thought-provoking, particularly in questioning the common belief that reflections primarily serve as a self-correction mechanism. However, certain experimental designs and interpretations could benefit from further clarification or control.

**Strengths:**

1. The paper addresses a novel and underexplored topic — the reflection behavior in reasoning models — and formulates two fundamental questions that are both insightful and significant to the field.

2. The authors conduct a comprehensive experimental study across a wide range of reasoning models and five mathematical benchmarks of varying difficulty, demonstrating both breadth and depth in evaluation.

3. The proposed LLM-based method for extracting intermediate answers and identifying reflection segments is well-motivated and technically sound. The validation of extraction accuracy through manual annotations further strengthens the reliability of the analysis.

4. The paper provides valuable insights into the distinction between confirmatory and corrective reflections and introduces a practical inference-time early-stopping strategy that effectively balances computation cost and performance.

**Weaknesses:**

1. When investigating the influence of reflection during training, the authors rely solely on supervised fine-tuning (SFT) to control reflective reasoning behavior. However, recent studies have shown that reinforcement learning (RL) can alter reasoning patterns in ways that differ substantially from SFT. It remains unclear whether the findings from the controlled SFT experiments would generalize to RL training. A discussion or empirical verification in this direction would be valuable.

2. According to Figure 5, models tend to exhibit more reflections on easier problems and fewer on difficult ones, suggesting a misalignment between reflection behavior and task difficulty. It would strengthen the paper if the authors could analyze specific cases or provide plausible explanations for this counterintuitive trend.

3. The paper reports a negative correlation between reflection amount and problem difficulty. However, in constructing training data for controlled reflection experiments, the authors only select samples with a reflection amount greater than or equal to six. This design choice might introduce a difficulty bias in the training distribution. The paper should clarify whether such bias exists and, if so, describe any measures taken to mitigate its impact.

4. The conclusion that “training on reflection-rich rollouts yields higher accuracy and longer generations” might be confounded by token-length differences. Longer rollouts naturally contain more tokens, which could encode richer information beyond reflection alone. Thus, the observed performance gains might partially stem from larger token exposure rather than the reflection mechanism itself. It would be helpful if the authors could control for token length — for instance, by comparing responses of similar length but differing reflection amounts — to more rigorously isolate the reflection effect.

**Questions:**

Please see weaknesses

---

> ### Author Response · Authors · 2025-11-15
> **Author response 1/2**
>
> Thank you for your insightful review. We address the concerns below.
>
> ### Weakness 1.
> _Relation to reinforcement learning_
>
> Thank you for pointing this out. In our paper, we briefly discuss the implications of our findings on RL training in the last part of Section 3.1 and Figure 8, where we compared the behaviour of two checkpoints before and after RL, and showed that the RL stage increases performance mainly by boosting the accuracy of the first try.
>
> To further investigate this, we conducted an RL training experiment, training a model initialized from Qwen2.5-Math-7B, which is a base model, with the DeepScaleR dataset [1] and GRPO algorithm. We collect rollouts on the five mathematical datasets studied in the paper for every 5 steps of RL training and analyze their first answer accuracy and final answer accuracy.
>
> As shown in Figure 16 of Appendix Section K in the updated PDF, the first-answer-accuracy closely matches the final answer accuracy as training progresses. This echoes with our findings that the performance of a reasoning model is mainly decided by its first answer accuracy, and the model’s performance boost from RL also mainly originates from the boost in first answer accuracy.
>
> [1] DeepScaleR: Surpassing O1-Preview with a 1.5B Model by Scaling RL. Michael Luo, Sijun Tan, Justin Wong, Xiaoxiang Shi, William Y. Tang, Manan Roongta, Colin Cai, Jeffrey Luo, Li Erran Li, Raluca Ada Popa, Ion Stoica. 2025.
>
> ### Weakness 2.
> _Further investigation on why easier problems have more reflections_
>
> Thank you for suggesting this. To further investigate this counterintuitive trend, where easier problems exhibit more reflections and harder problems exhibit fewer, we take a closer look at the rollouts of Qwen3-8B on Math500 (a relatively easy dataset) and AIME2025 (a harder dataset). We report the average appearing position of the first candidate answer, average total length, average number of reflections, and average lengths of reflections of the two datasets in the table below.
>
> From the table, we can see that
>
> 1) Math500 has an overall shorter total length, as well as a much earlier first candidate answer appearing position (2522 compared to 12254 of AIME2025). This is expected as the problems in AIME2025 are harder and require more reasoning to get the first candidate answer.
>
> 2) The model spends more tokens on reflection on AIME2025 than Math500 (Math500: 4867-2522=2345 tokens, AIME2025: 15959-12254=3705 tokens).
>
> 3) However, since problems in AIME2025 are harder, the reflections are usually more complicated than those of Math500; as a result, the average length of reflections on AIME2025 (1538 tokens) is much longer than that of Math500 (418 tokens)
> These observations provide a clearer picture of why the counterintuitive trend occurs: harder problems require more sophisticated reflections, thus consuming more tokens, and as a result, the number of reflections the model produces is usually less than that on easier problems.
>
> |  | Math500 | AIME2025 |
> | --- | --- | --- |
> | First Candidate Position (#tokens) | 2522 | 12254 |
> | Total Length (#tokens) | 4867 | 15959 |
> | Average Number of Reflections | 5.71 | 2.47 |
> | Average Lengths of Reflections (#tokens) | 418 | 1538 |
>
> ### Weakness 3.
> _SFT data selection bias_
>
> Thank you for this insightful observation. We agree that limiting to only including rollouts with greater than or equal to six candidate answers introduces a selection bias where easier problems are more likely to be selected. Since we want to have controlled experiments where the set of problems should be the same, and the only changing factor is the number of reflections, this selection bias is hard to avoid. In order to address the concern about this bias, we conduct an additional experiment, where we relax the criteria and select all samples with greater than or equal to three candidate answers (which is 62.7% of all samples). This relaxation mitigates the selection bias so that harder problems are more likely to be included in the SFT dataset than in the previous setting. Using these newly curated cut-at-{i} (i=1,2,3) datasets, we perform SFT with Qwen2.5-7B-Instruct. Results are reported in the table below. We can see that the pattern persists when we relaxed the selection condition, suggesting that the observed pattern is not due to the fact that we were previously biased toward selecting easy problems in the controlled SFT dataset.
>
> | Qwen2.5-7B-Instruct | First answer accuracy | Final accuracy | First answer relative position | Total length |
> | --- | --- | --- | --- | --- |
> | cut-at-1 | 43.3 | 45.5 | 0.89 | 7763 |
> | cut-at-2 | 44.3 | 47.5 | 0.65 | 9283 |
> | cut-at-3 | 46.5 | 48.1 | 0.45 | 12761 |

---

> ### Author Response · Authors · 2025-11-15
> **Author response 2/2**
>
> ### Weakness 4.
> _Concerns on token exposure and why training with reflection-rich rollouts help performance_
>
> Thank you for pointing this out. In our SFT data curation procedure, we have taken into account that longer rollouts have more training tokens, thus may teach the model more information. To eliminate this issue, as described in Section 3.1, we downsampled for larger i in cut-at-{i} dataset, so that all of them have 28 million training tokens. Consequently, for larger i, the dataset actually has fewer problems than that of smaller i, however, in our controlled SFT experiments we see that it still yields higher performance when trained on larger i datasets, even with less number of training rollouts.
>
> In the paper, we made one hypothesis about why “training on reflection-rich rollouts yields higher accuracy and longer generations”. We mentioned that this could be because reflection-rich rollouts have more diverse problem-solving approaches, presenting more than one method to solve the same problem, and that models trained on such data therefore generalize better. To further investigate this hypothesis, we conducted analysis on the curated SFT datasets (Section 3.1). To assess whether datasets truncated at later positions include extra useful information, we compare two responses to the same problem: one from dataset cut-at-{i} and the other from dataset cut-at-{i+1}, where the latter contains one more piece of reflection. We evaluate whether this added text introduces any new mathematical concepts or problem-solving approaches by prompting an LLM (gpt-oss-120b) (see the updated Section I in the appendix for the prompt). For all problems, we then compute the percentage for which cut-at-{i+1} contributes additional concepts/approaches beyond cut-at-{i}. The statistics are shown in the table below.
>
> |  | Percentage of problems with diverse information |
> | --- | --- |
> | From cut-at-1 to cut-at-2 | 22.6 |
> | From cut-at-2 to cut-at-3 | 16.8 |
> | From cut-at-3 to cut-at-4 | 15.6 |
> | From cut-at-4 to cut-at-5 | 16.4 |
> | From cut-at-5 to cut-at-6 | 15.2 |
>
> From the table, we can see that while some additional reflections are simple repetitive calculations, a reasonable number of the problems see additional problem-solving approaches in the longer rollout. These additional approaches to the same problem increase the diversity of the rollout and may help the model generalize better.
> We also provide two cases in the updated PDF, showing when the rollout with more reflections provides additional problem-solving approaches (Figure 14) and when the rollout with more reflections repeats previous calculations (Figure 15).
>
> We hope the discussion and additional material can help address your concerns, and we are happy to clarify any further questions.

---

> ### Author Response · Authors · 2025-11-27
> **Follow-up on Discussion for Submission "First Try Matters: Revisiting the Role of Reflection in Reasoning Models"**
>
> Dear Reviewer hGc3,
>
> I hope this message finds you well.
>
> We would like to kindly follow up regarding our response to your concerns and questions. We truly appreciate the time and effort you have already invested in reviewing our work. Since your review plays an important role in the final decision, we wanted to check whether you might have any further comments or questions after reading our response.
>
> We understand this is a busy period and apologize for the interruption—thank you again for your valuable feedback and for helping improve our work.
>
> Warm regards, Authors

---

### Official Review · Reviewer_LMU1 · 2025-10-31

**Soundness:** 3
**Presentation:** 3
**Contribution:** 3
**Rating:** 6
**Confidence:** 3

**Summary:**

This paper do a analysis on CoT rollouts and find they mostly confirm the first answer rather than fix errors. Then authors proposed a question-aware early-stopping method that detects candidate answers to truncate reflections, cutting tokens by about 24.5% with about a 2.9% accuracy drop.

**Strengths:**

1. very interesting findings on the large-scale cot first-try answer correctness
2. the proposed method is straightforward and intuitive

**Weaknesses:**

1. all analysis and the efficiency method are evaluated only on five mathematical datasets, so generalization to other domains is unclear
2. early stopping cuts tokens but incurs non-trivial accuracy drops on hard sets
3. CAD and QRC are trained on LLM-generated labels and depend on a hand-crafted candidate-extraction prompt, which may be sensitive to formatting, symbols, and units
4. the CAD+QRC early-stopping demos are shown on Qwen3-8B only; portability to other reasoning models is not demonstrated.

**Questions:**

N/A

---

> ### Author Response · Authors · 2025-11-15
> **Author response 1/2**
>
> Thank you for your insightful review. We address the concerns below.
>
> ### Weakness 1.
> > all analysis and the efficiency method are evaluated only on five mathematical datasets, so generalization to other domains is unclear
>
> Thank you for pointing this out. To see how our findings generalize to other domains, we perform analysis on three MMLU Tasks: Formal Logic, High School Statistics, and Economics. For each problem, we collect reasoning rollouts from Qwen3-8B, Qwen3-14B, and DeepSeek-R1-Distill-Llama-8B. We then extract the positions of candidate answers following the same procedure in the paper (see Section J in the updated PDF for the extraction prompt for MMLU).
>
> In the table below, we report the first-answer accuracy, the final-answer accuracy, and the relative position of the first answer. We can see that on these non-mathematical tasks, the models exhibit a similar pattern: the first-answer accuracy closely matches the final-answer accuracy; meanwhile, the relative position of the first answer is early.
>
> To summarize, the experiments on three other domains show a similar pattern as in mathematical domains, suggesting that the reasoning process of current LLMs is largely determined by their first answer accuracy, while later reflections hardly change the answer.
>
> | Qwen3-8B | Formal Logic | High School Statistics | Economics |
> | --- | --- | --- | --- |
> | First answer accuracy | 94.4 | 91.7 | 79.8 |
> | Final answer accuracy | 96.8 | 94.4 | 80.7 |
> | First answer relative position | 0.47 | 0.51 | 0.54 |
>
> | Qwen3-14B | Formal Logic | High School Statistics | Economics |
> | --- | --- | --- | --- |
> | First answer accuracy | 96.8 | 93.1 | 78.9 |
> | Final answer accuracy | 97.6 | 94.4 | 80.7 |
> | First answer relative position | 0.51 | 0.56 | 0.59 |
>
> | DeepSeek-R1-Distill-Llama-8B | Formal Logic | High School Statistics | Economics |
> | --- | --- | --- | --- |
> | First answer accuracy | 64.3 | 77.8 | 45.6 |
> | Final answer accuracy | 66.7 | 79.2 | 48.2 |
> | First answer relative position | 0.66 | 0.69 | 0.74 |
>
> ### Weakness 2.
> > early stopping cuts tokens but incurs non-trivial accuracy drops on hard sets
>
> We agree that the early stopping mechanism introduced by CAD and QRC may cause an accuracy drop. However, we argue that the CAD and QRC provide a **controllable balance** between token usage and performance. As shown in the bottom left corner of Figure 9, when the early stopping criteria are strict (achieved by adjusting the threshold), CAD and QRC achieve a 12% token usage drop, while the performance drops only 1%.
>
> Additionally, we show that CAD and QRC do not incur more performance drop on harder problems by showing their performance on AIME2025, which is considered the hardest among the five studied datasets. Due to the space limit here, please refer to Figure 17 (c) and (d) in the updated pdf for the results. Comparing Figure 17 (c) with (e) and (d) with (f), we can see that on AIME2025, CAD and QRC do not introduce more performance drop than average, suggesting their robustness to problem difficulties. Moreover, we also see that in some threshold configurations of CAD and QRC in Figure 17, we can **achieve some performance improvement by early stopping the thinking process**, suggesting that excessive reflections may have negative effects on the performance.

---

> ### Author Response · Authors · 2025-11-15
> **Author response 2/2**
>
> ### Weakness 3.
> > CAD and QRC are trained on LLM-generated labels and depend on a hand-crafted candidate-extraction prompt, which may be sensitive to formatting, symbols, and units
>
> Thank you for pointing this out. While the labels used for training CAD and QRC are LLM generated, we provided two lines of evidence to show that such labels are robust: 1) In Figure 2, we show that the LLM extraction procedure is robust to change of LLM model and change of prompts; 2) We provide human evaluation of the LLM labeled candidate positions in Section C in Appendix, showing that human participants mostly agree with the LLM annotation. Moreover, we want to point out that the task assigned to CAD is simple: it takes in the problem statement and **one** sentence from the reasoning trace, and decides whether the particular sentence answers the problem. To do this task well, the CAD only needs to understand what quantity the problem is asking for, and whether the sentence contains it, without requiring any knowledge about the problem itself or judging the correctness of the problem, which is also the reason we use a smaller model for initializing CAD (Qwen3-1.7B).
>
> To demonstrate that CAD and QRC can transfer well, we conduct the following experiments. We apply CAD and QRC (which are trained using LLM-annotated Qwen3-8B rollouts) to two other models: DeepSeek-R1-Distill-Llama-8B and Qwen3-14B. In addition to testing these two models on the five mathematical datasets studied in the paper, we test on three more recent mathematical datasets, CMIMC 2025 (March 2025), HMMT 2025 (February 2025), and BRUMO 2025 (April 2025), to show the robustness of CAD and QRC to problems that have varying formatting, symbols, and units.
>
> Due to space limitations, we show the results of Qwen3-14B on the three recent datasets. Please refer to Appendix Section L for a detailed table of different models on different datasets.
>
> Accuracy Percentage-Point Change
>
> | CAD \ QRC | 0 | 0.01 | 0.05 | 0.1 | 0.5 | 1.0 |
> | --- | --- | --- | --- | --- | --- | --- |
> | **0.15** | -3.0 | -2.6 | -7.0 | -8.0 | -9.5 | -9.5 |
> | **0.30** | +1.9 | +1.9 | -1.6 | -3.9 | -4.2 | -4.2 |
> | **0.50** | +0.7 | +0.7 | -0.3 | -2.3 | -2.7 | -2.7 |
>
> Token Reduction (%)
>
> | CAD \ QRC | 0 | 0.01 | 0.05 | 0.1 | 0.5 | 1.0 |
> | --- | --- | --- | --- | --- | --- | --- |
> | **0.15** | -21.6 | -21.9 | -30.6 | -33.8 | -36.8 | -37.0 |
> | **0.30** | -13.5 | -13.8 | -22.4 | -27.1 | -30.7 | -30.8 |
> | **0.50** | -5.6 | -6.0 | -13.6 | -17.0 | -18.7 | -18.7 |
>
> From the table above, we can see that the CAD and QRC trained using annotated rollouts of Qwen3-8B can transfer well when applied to Qwen3-14B on the most recent mathematical datasets. Suggesting its robustness to model and problem formatting. Moreover, we also see that in some threshold configurations of CAD and QRC (e.g. 0.5 and 0), we achieve a 1.9% performance increase while reducing 13.5% of the tokens by early stopping. Suggesting that excessive reflections may even have negative effects on the model’s reasoning performance.
>
> ### Weakness 4.
> > the CAD+QRC early-stopping demos are shown on Qwen3-8B only; portability to other reasoning models is not demonstrated.
>
> Please see our response to Weakness 3.
>
> We hope the discussion and additional material can help address your concerns, and we are happy to clarify any further questions.

---

> ### Author Response · Authors · 2025-11-27
> **Follow-up on Discussion for Submission "First Try Matters: Revisiting the Role of Reflection in Reasoning Models"**
>
> Dear Reviewer LMU1,
>
> I hope this message finds you well.
>
> We would like to kindly follow up regarding our response to your concerns and questions. We truly appreciate the time and effort you have already invested in reviewing our work. Since your review plays an important role in the final decision, we wanted to check whether you might have any further comments or questions after reading our response.
>
> We understand this is a busy period and apologize for the interruption—thank you again for your valuable feedback and for helping improve our work.
>
> Warm regards, Authors

---

### Official Review · Reviewer_wnXZ · 2025-11-01

**Soundness:** 2
**Presentation:** 3
**Contribution:** 2
**Rating:** 4
**Confidence:** 4

**Summary:**

This paper challenges the common assumption that "reflection" in large language models—the reasoning steps that occur after a model has already produced a candidate answer—serves to correct mistakes.
The authors' large-scale analysis reveals two key findings:
1. Inference-Time Reflections are Confirmatory, Not Corrective: When models "reflect," they overwhelmingly just confirm their initial answer, whether it was right or wrong. True self-correction (changing a wrong answer to a right one) is extremely rare (less than 2% of the time).
2. Training-Time Reflections Boost First-Try Accuracy: While reflections are ineffective for correction during inference, training on data that includes long reflections is beneficial. However, the benefit does not come from teaching the model to fix its mistakes. Instead, it significantly improves the model's ability to get the answer right on its very first try.
Based on these findings, the authors propose an adaptive early-stopping method. By cutting off these unnecessary and computationally expensive reflection steps during inference, they can reduce token usage by 24.5% with only a minimal 2.9% drop in accuracy.

**Strengths:**

1. Significant and Counter-intuitive Research Question:
This paper challenges a widely held assumption in the field of LLM reasoning that "reflection" (the steps after generating an answer) is an active, human-like self-correction mechanism. The authors question this "reflection = correction" intuition, which serves as a novel and high-impact entry point. In a community where "longer chain-of-thought = better reasoning" is a common belief, this paper's focus is especially timely and important.
2. Robust Large-Scale Empirical Analysis:
The paper's core argument is not based on speculation or anecdotes but on solid quantitative data. The authors systematically analyzed 8 different advanced reasoning models (covering both RL and SFT) across 5 math benchmarks. The quantitative findings such as "over 90% of reflections are confirmatory" and "less than 2% are truly corrective" ($F \rightarrow T$)—provide exceptionally strong evidence for the paper's claims, demonstrating that this is a general phenomenon across models and datasets.
3. Clever Experimental Design:
  - "Candidate Answer Extractor": The authors designed an ingenious methodology to clearly partition a model's thought process (rollout) into "forward reasoning" (up to the first candidate answer) and "reflective reasoning" (everything after). This is the key methodological contribution that enables the entire analysis.
  - "Cut-at-i" Supervised Fine-Tuning (SFT) Experiment: This experiment is particularly outstanding. By controlling the total training token budget while varying the ratio of "reflection length" to "number of problems," the authors successfully isolated the true role of reflection data in training. This experiment powerfully demonstrates that the value of reflection data is not in teaching the model "how to correct" but rather in significantly improving its "first-try accuracy" by providing diverse reasoning paths.

**Weaknesses:**

1. Potential Data Contamination in Evaluation

The paper uses Math500 and other datasets to evaluate reflection behavior, but Math500 (published in 2021) likely overlaps with the training data of the evaluated models. This contamination could artificially inflate first-try accuracy, making reflections appear less useful than they actually are. The analysis would be more convincing if conducted primarily on temporal holdout sets (e.g., AIME2024/2025) or with explicit discussion of data leakage impact.

2. Biased Training Data in SFT Experiments

The SFT experiments (Section 3.1) only use rollouts with correct final answers, which fundamentally biases the training distribution toward first-try success patterns rather than error correction patterns. This design prevents models from learning self-correction behavior and directly causes the observed outcome that "reflections improve first-try accuracy."
Furthermore, the paper evaluates improvements using absolute values rather than relative growth rates. Calculating from Figure 7, the reflection improvement shows a 33% relative increase (0.9%→1.2%) compared to only 19% for first-try accuracy (18.5%→22.0%), suggesting reflections may be more impactful than claimed.

3. Unverified "Diverse Reasoning Paths" Hypothesis

The paper hypothesizes that reflection-rich rollouts improve generalization by exposing diverse reasoning paths (Section 3.1), but provides no experimental validation. This remains speculation without analyses such as: measuring reasoning path diversity, ablation studies comparing single long rollouts vs. multiple diverse rollouts, or visualization of intermediate reasoning representations.

4. Missing Computational Efficiency Analysis

The proposed early-stopping method claims efficiency gains but only reports token reduction (24.5%). However, the method requires running two additional models (CAD and QRC), with CAD being invoked serially multiple times during generation. The paper provides no measurements of total FLOPs, wall-clock latency, throughput, or memory overhead. Without these comprehensive efficiency benchmarks, it is unclear whether the method achieves genuine computational savings or if the overhead from auxiliary models offsets the token reduction benefits.

**Questions:**

1. The efficiency analysis relies on token reduction. As token count can be an imperfect proxy for computational cost (e.g., prefill vs. decoding), could the authors provide an analysis in terms of FLOPs to more accurately quantify the computational savings?
2. The paper posits that reflections improve first-try accuracy by exposing the model to 'diverse reasoning paths.' This is a key hypothesis. Could the authors provide more direct evidence to support this claim, perhaps by quantifying this 'diversity' and correlating it with the downstream performance improvements?

---

> ### Author Response · Authors · 2025-11-15
> **Author response 1/3**
>
> Thank you for your insightful review. We address the concerns below.
>
> ### Weakness 1.
> > Potential Data Contamination in Evaluation
>
> We agree that potential data contamination is an important concern when evaluating contemporary LLMs.
>
> We would like to clarify that our analysis is not solely based on Math500 (2021). As summarized in Section 2.1, it draws on multiple datasets from different years.  Several of these datasets are recent and therefore unlikely to appear in model pretraining corpora, including AMC (2023), OlympiadBench (2024), AIME24 (2024), and AIME25 (2025). These datasets were released within the past two years, with AIME25 appearing this year, which makes contamination less likely.
>
> As shown in Appendix D and Appendix G, which provide dataset-level breakdowns and additional details, the trends in reflection behavior remain consistent across all datasets. This indicates that the potential overlap in Math500 does not affect our conclusions.
>
> To further investigate the concern of data leakage, we additionally evaluate **three recent mathematical competitions that took place in 2025**: CMIMC 2025 (March 2025), HMMT 2025 (February 2025), and BRUMO 2025 (April 2025) [1]. These datasets are much less likely to suffer from data contamination, and we hope their results, as shown in the table below, help address the concern.
>
> | Qwen3-8B | CMIMC | HMMT | BRUMO |
> | --- | --- | --- | --- |
> | First answer accuracy | 41.9 | 48.3 | 67.5 |
> | Final answer accuracy | 43.1 | 48.3 | 69.2 |
> | First answer relative position | 0.67 | 0.64 | 0.52 |
>
> | Qwen3-14B | CMIMC | HMMT | BRUMO |
> | --- | --- | --- | --- |
> | First answer accuracy | 50.6 | 55.8 | 77.5 |
> | Final answer accuracy | 51.9 | 57.5 | 79.2 |
> | First answer relative position | 0.63 | 0.65 | 0.52 |
>
> | DeepSeek-R1-Distill-Llama-8B | CMIMC | HMMT | BRUMO |
> | --- | --- | --- | --- |
> | First answer accuracy | 14.4 | 15.0 | 34.2 |
> | Final answer accuracy | 15.0 | 16.7 | 36.7 |
> | First answer relative position | 0.77 | 0.81 | 0.69 |
>
> From the table, we see that across all three 2025 datasets, the patterns revealed in the paper remain unchanged: reflections are largely confirmatory and seldom modify the model’s initial answer, with first answer accuracy closely matching the final answer accuracy, while appearing at early positions.
>
> In summary, although Math500 may partially overlap with pretraining data, **our results on recent and contamination-resistant datasets confirm that the main conclusion of the paper is robust: the first try matters substantially for current reasoning models**.
>
> [1] MathArena: Evaluating LLMs on Uncontaminated Math Competitions, 2025 May, Neurips 2025

---

> ### Author Response · Authors · 2025-11-15
> **Author response 2/3**
>
> ### Weakness 2.
> > Biased Training Data in SFT Experiments
>
> 1. On the data selection bias
>
> Thank you for the observation.
>
> > only use rollouts with correct final answers, which fundamentally biases the training distribution toward first-try success patterns rather than error correction patterns
>
> We want to point out the misunderstanding here: the "error correction patterns" would also lead to a correct final answer, so filtering the training data by correctness wouldn’t affect the model learning the error correction patterns. Moreover, it is standard for SFT to use rollouts that have a correct final answer.
>
> We also share the concern that it may be hard for the model to learn error correction if most of the training rollouts contain mostly confirmatory reflections. To investigate this, we conducted the experiments in Section 3.2 of the paper, where we fix the same set of problems for training and vary the percentage of problems for which we use a rollout whose reflections are all confirmatory versus a rollout that contains “F→T” transitions. As shown in Table 1 in the paper, **varying the amount of corrective reflections in the training data does not have a noticeable impact on the model’s performance**, suggesting that the corrective patterns can hardly be learned through SFT.
>
> 2. On the use of absolute vs. relative improvement metrics
>
> We appreciate the reviewer’s point about relative improvement. However, we believe absolute metrics are the more appropriate and widely used measure for two reasons:
>
> (a) Field practice: Prior work on reasoning and reflection consistently reports absolute accuracy gains, not relative ratios [1, 2, 3, 4, 5]. We follow this standard to ensure consistency and interpretability.
>
> (b) Denominator effect: Using relative gains can be misleading when the denominator is small (e.g., 0.9% → 1.2%). A tiny absolute change can appear artificially large in relative terms. In contrast, the first answer accuracy operates at a much larger base rate (e.g., ~20%), so relative ratios are not comparable across metrics with dramatically different scales. This asymmetry creates the impression that reflection improvements are “larger,” even when their absolute contributions remain marginal in practical impact.
>
> In our setting, the key quantity of interest is actual accuracy gain delivered to end performance, and on that axis, the first answer accuracy is the dominating factor.
>
> [1] DeepSeek-R1: Incentivizing Reasoning Capability in LLMs via Reinforcement Learning
>
> [2] Demystifying Long Chain-of-Thought Reasoning
>
> [3] Thoughts are all over the place: On the underthinking of o1-like llms
>
> [4] Revisiting the Test-Time Scaling of o1-like Models: Do they Truly Possess Test-Time Scaling Capabilities?
>
> [5] Prorl: Prolonged reinforcement learning expands reasoning boundaries in large language models

---

> ### Author Response · Authors · 2025-11-15
> **Author response 3/3**
>
> ### Weakness 3.
> > Unverified "Diverse Reasoning Paths" Hypothesis
>
> Thank you for pointing this out. From the experiment results in Figure 6, we see that training with rollouts that have more reflections yields better performance. The datasets curated in Section 3.1 are: 1) from the same teacher model, 2) on the same set of problems (for larger i in cut-at-{i} dataset, the problems are a subset of those of the smaller i, so that all datasets have the same total amount of training tokens). Given that these variables are controlled, the only differing factor is the number of reflections in the training rollout; thus, we made the hypothesis in the paper that the performance improvements may originate from rollouts with more reflections, which are more diverse and provide multiple approaches to solve the problem.
>
> To show that rollouts with more reflections indeed have such diversity, we conducted further analysis on the curated SFT datasets (Section 3.1). To assess whether datasets truncated at later positions include extra useful information, we compare two responses to the same problem: one from dataset cut-at-{i} and the other from dataset cut-at-{i+1}, where the latter contains one more piece of reflection. We evaluate whether this added text introduces any new mathematical concepts or problem-solving approaches by prompting an LLM (gpt-oss-120b) (see the updated Section I in the appendix for the prompt). For all problems, we then compute the percentage of problems for which cut-at-{i+1} contributes additional concepts/approaches beyond cut-at-{i}. The statistics are shown in the table below.
>
> |  | Percentage of problems with additional information |
> | --- | --- |
> | From cut-at-1 to cut-at-2 | 22.6 |
> | From cut-at-2 to cut-at-3 | 16.8 |
> | From cut-at-3 to cut-at-4 | 15.6 |
> | From cut-at-4 to cut-at-5 | 16.4 |
> | From cut-at-5 to cut-at-6 | 15.2 |
>
> From the table, we can see that while some additional reflections are simple repetitive calculations, a reasonable number of the problems see additional problem-solving approaches in the longer rollout. These additional approaches to the same problem increase the diversity of the rollout and may help the model generalize better.
> We also provide two cases in the updated PDF, showing when the rollout with more reflections provides additional problem-solving approaches (Figure 14) and when the rollout with more reflections repeats previous calculations (Figure 15).
>
> ### Weakness 4.
> > Missing Computational Efficiency Analysis
>
> Thank you for raising this important point. Our method introduces two additional components: (1) a single forward pass from QRC per problem to determine the reflection budget, and (2) lightweight CAD evaluations triggered only at linebreak tokens during generation.
>
> Both modules operate on smaller models (1.7B parameters) than the main reasoning model and are invoked far less frequently. QRC is called only once per problem, and CAD is triggered only at linebreak tokens. In contrast, the base model performs thousands of forward passes for the long reasoning sequence. As a result, the additional computation is negligible compared with the decoding cost it saves.
>
> To quantify the actual computational cost, we conducted an efficiency study. We randomly sampled 100 problems from the five mathematical benchmarks and ran all experiments sequentially on a single NVIDIA A800 GPU using Qwen3-8B. We report average generation speed (tokens/s) and estimated FLOPs per problem.
>
> |  | Wall-clock (seconds per problem) | Speed (#tok/s) | FLOPs per problem estimation |
> | --- | --- | --- | --- |
> | Baseline | 486.8 | 18.40 | 1.28*10^14 |
> | +CAD(thres=0.5),QRC(thres=0.5) | 242.2 | 18.05 | 6.54*10^13 |
>
> We find that the overhead from CAD and QRC is minimal. The average generation speed decreases only slightly, from 18.40 tok/s to 18.05 tok/s (–1.9%). In contrast, early stopping reduces the total number of generated thinking tokens substantially (Figure 9: 12.0% to 40.9% across datasets). Together, these effects yield a 50.24% end-to-end wall-clock time reduction.
>
> Moreover, the thresholds used by CAD and QRC are simple hyperparameters that can be adjusted as needed, allowing the system to flexibly tune the stopping criterion based on the desired balance between accuracy and efficiency.
>
> Overall, these results show that the auxiliary models contribute negligible overhead relative to the tokens saved, and the system delivers **genuine end-to-end computational gains** rather than merely shifting the cost elsewhere.
>
> ### Question 1.
> _Efficiency analysis of proposed token reduction technique_
>
> Please see our response to Weakness 4.
>
> ### Question 2.
> _Evidence on the hypothesis of "training with diverse reasoning paths improve model performance"_
>
> Please see our response to Weakness 3.
>
> We hope the discussion and additional material can help address your concerns, and we are happy to clarify any further questions.

---

> ### Author Response · Authors · 2025-11-25
> **Follow-up on Discussion for Submission "First Try Matters: Revisiting the Role of Reflection in Reasoning Models"**
>
> Dear Reviewer wnXZ,
>
> I hope this message finds you well.
>
> We would like to kindly follow up regarding our response to your concerns and questions. We truly appreciate the time and effort you have already invested in reviewing our work. Since your review plays an important role in the final decision, we wanted to check whether you might have any further comments or questions after reading our response.
>
> We understand this is a busy period and apologize for the interruption—thank you again for your valuable feedback and for helping improve our work.
>
> Warm regards,
> Authors

---

### Official Review · Reviewer_hGUM · 2025-11-01

**Soundness:** 3
**Presentation:** 3
**Contribution:** 3
**Rating:** 6
**Confidence:** 3

**Summary:**

This paper investigates the role of reflective reasoning in large language models, showing that most reflection steps reinforce an initial answer rather than correcting it. The authors further demonstrate that training on reflection-rich rollouts primarily improves first-attempt accuracy, and they propose an early-stopping strategy that limits unnecessary reasoning while largely preserving model performance.

**Strengths:**

1. The paper reveals an interesting finding: training models on rollouts with more reflection steps primarily improves their ability to answer correctly on the first attempt, rather than enhancing their capability to correct mistakes through reflection.

2. The presentation is clear and well-structured.

**Weaknesses:**

1. It would be better to expand the experiment in Section 3.1 to larger models, such as a 14B version.

2. Regarding the statement *“One possible explanation is that richer reflections expose the model to diverse problem-solving approaches, improving generalization and boosting initial answer quality rather than simply correcting mistakes.”*, it would be valuable to provide further discussion or empirical evidence to support this point. In addition, exploring this phenomenon from an interpretability perspective and examining the actual training dynamics would make the paper more comprehensive and insightful.

**Questions:**

Please see the weaknesses above.

---

> ### Author Response · Authors · 2025-11-15
>
> Thank you for your insightful review. We address the concerns below.
>
> ### Weakness 1.
> > expand the experiment in Section 3.1 to larger models, such as a 14B version
>
> Thank you for the suggestion. To evaluate the impact of reflection amount on SFT performance on larger models, we train a Qwen2.5-14B-Instruct following the same experiment settings as shown in Figure 6. We use the dataset curated in Section 3.1 for training and test the trained models on the five mathematical datasets in the paper (AIME2024, AIME2025, AMC, Olympiad Bench, and Math500). We report the average accuracy and average length of the trained models in the table below. We can see that the controlled SFT experiments on a larger 14B model show a similar pattern as in Figure 6, where models trained on rollouts that have more reflections demonstrate higher accuracy and longer response length.
>
> | Qwen2.5-14B-Instruct | First answer accuracy | Final accuracy | First answer relative position | Total length |
> | --- | --- | --- | --- | --- |
> | cut-at-1 | 49.8 | 51.6 | 0.97 | 6170 |
> | cut-at-2 | 51.4 | 53.2 | 0.79 | 7082 |
> | cut-at-3 | 52.1 | 54.6 | 0.66 | 8922 |
> | cut-at-4 | 50.9 | 53.5 | 0.61 | 9725 |
> | cut-at-5 | 52.8 | 55.4 | 0.50 | 12020 |
> | cut-at-6 | 53.2 | 55.8 | 0.44 | 13584 |
>
> ### Weakness 2.
> _Further discussion on the hypothesis that "richer reflections expose the model to diverse problem-solving approaches, improving generalization and boosting initial answer quality rather than simply correcting mistakes"_
>
> Thank you for pointing this out. To provide additional empirical evidence that training data with more reflections contains more diverse problem-solving approaches, and that models trained on such data therefore generalize better, we conducted further analysis on the curated SFT datasets (Section 3.1). To assess whether datasets truncated at later positions include extra useful information, we compare two responses to the same problem: one from dataset cut-at-{i} and the other from dataset cut-at-{i+1}, where the latter contains one more piece of reflection. We evaluate whether this added text introduces any new mathematical concepts or problem-solving approaches by prompting an LLM (gpt-oss-120b) (see the updated Section I in the appendix for the prompt). For all problems, we then compute the percentage for which cut-at-{i+1} contributes additional concepts/approaches beyond cut-at-{i}. The statistics are shown in the table below.
>
> |  | Percentage of problems with additional information |
> | --- | --- |
> | From cut-at-1 to cut-at-2 | 22.6 |
> | From cut-at-2 to cut-at-3 | 16.8 |
> | From cut-at-3 to cut-at-4 | 15.6 |
> | From cut-at-4 to cut-at-5 | 16.4 |
> | From cut-at-5 to cut-at-6 | 15.2 |
>
> From the table, we can see that while some additional reflections are simple repetitive calculations, a reasonable number of the problems see additional problem-solving approaches in the longer rollout. These additional approaches to the same problem increase the diversity of the rollout and may help the model generalize better.
> We also provide two cases in the updated PDF, showing when the rollout with more reflections provides additional problem-solving approaches (Figure 14) and when the rollout with more reflections repeats previous calculations (Figure 15).
>
> We hope the discussion and additional material can help address your concerns, and we are happy to clarify any further questions.

---

> ### Author Response · Authors · 2025-11-27
> **Follow-up on Discussion for Submission "First Try Matters: Revisiting the Role of Reflection in Reasoning Models"**
>
> Dear Reviewer hGUM,
>
> I hope this message finds you well.
>
> We would like to kindly follow up regarding our response to your concerns and questions. We truly appreciate the time and effort you have already invested in reviewing our work. Since your review plays an important role in the final decision, we wanted to check whether you might have any further comments or questions after reading our response.
>
> We understand this is a busy period and apologize for the interruption—thank you again for your valuable feedback and for helping improve our work.
>
> Warm regards, Authors

---

### Author Response · Authors · 2025-11-23

Dear Reviewers,

Thank you again for spending time reviewing and providing high-quality reviews to our paper. We have provided detailed responses to address the concerns. We would greatly appreciate your feedback and evaluation of the updated version of the paper, and we are happy to provide any further clarifications. Looking forward to your reply.

Best regards,
Authors

---

> ### Author Response · Authors · 2025-11-24
>
> We sincerely thank all reviewers for their thoughtful and constructive feedback. In response, we conducted substantial additional experiments, expanded analyses, and clarified key methodological choices. Below is a consolidated summary of the major updates and rebuttal points across all reviewers.
>
> ---
>
> ## **Key Additions and Clarifications**
>
> - **Added controlled SFT experiments on larger model (14B).** (_Reviewer hGUM_) We show that the pattern "training with more reflections consistently improves first-try and final accuracy" shown in the paper generalizes to larger models.
> - **Added quantitative diversity analysis.** (_Reviewer hGUM, wnZX, hGc3_) To support the hypothesis that _"One possible explanation is that richer reflections expose the model to diverse problem-solving approaches, improving generalization and boosting initial answer quality rather than simply correcting mistakes."_, we performed quantitative analysis of the diversity of rollouts with more/fewer reflections and showed that rollouts with more reflections have more diversity, as they provide different approaches to solving the same problem.
> - **Added evaluation on three recent 2025 math competitions.** (_Reviewer wnXZ_) To address the concern about math data contamination in recent models, we added evaluation on three most recent 2025 math competitions (CMIMC, HMMT, BRUMO), showing that the patterns are consistent with the main body of the paper.
> - **Added FLOPs and wall-clock measurements analysis.** (_Reviewer wnXZ_) To show the proposed early-stopping generation with CAD+QRC introduces minimal overhead, we added FLOPs and wall-clock measurements. Results showed that they introduce minimal overhead and can reduce wall-clock generation time by 50.24%.
> - **Added generalization results on non-math MMLU tasks.** (_Reviewer LMU1_) To show the analysis on mathematical problems also generalizes to other domains, we conduct analysis on three non-math MMLU tasks (Formal Logic, Statistics, Economics), and showed that the patterns are consistent with mathematical tasks.
> - **Demonstrated that CAD+QRC transfer across models.** (_Reviewer LMU1_) To analyze the generalizability of CAD+QRC across models and problems, we added experiments to test them on different models and datasets, and showed that CAD and QRC trained with one model and dataset can generalize well to other models and datasets.
> - **Added RL training experiment.** (_Reviewer hGc3_) To investigate how the findings in the paper might relate to RL training of reasoning models, we performed analysis on intermediate rollouts during RL training and showed that during RL the performance improvements mainly come from first-answer accuracy, echoing the main finding of the paper.
> - **Provided more analysis on reflection and difficulty mismatch.** (_Reviewer hGc3_) We performed additional analysis to investigate the counterintuitive trend where easier problems often exhibit more reflections. We showed that this is related to the complexity of the problem and the average length of reflections for difficult/hard problems.
> - **Added additional SFT experiments under a relaxed (greater than 3 reflections) selection threshold.** (_Reviewer hGc3_) To mitigate the selection bias in data curation of the controlled SFT experiments, we performed additional experiments by relaxing the selection criteria reducing the bias. Results show that the trends persist in this setting.
>
> We also updated details of the additional experiments in the updated appendix of the paper. We kindly request reviewers to respond to our updated material, and we are happy to have any further discussions.

---

### Meta-Review · Area_Chair_jy4R · 2026-01-06

**Summary:**

This paper challenges the assumption that LLM self-correction effectively rectifies errors, revealing that inference-time reflection predominantly confirms initial answers rather than correcting them. The authors demonstrate that while training on CoT data improves performance, the gains stem from enhancing first-attempt accuracy rather than post-hoc correction capability. Consequently, they propose an Adaptive Early-Stopping method that reduces token consumption by 24.5% with only a marginal (2.9%) sacrifice in accuracy. This framework provides a more efficient approach to managing the trade-off between computational cost and model performance in reasoning tasks.

**Reviewer Concerns:**

Addressed Concerns
1. Experimental Rigor and Scope (Reviewers hGc3, wnXZ, LMU1, hGUM): Reviewers initially raised concerns regarding potential data contamination in benchmarks and bias in SFT training data. Additionally, the lack of experiments on mid-sized models (e.g., 14B parameters) was noted. During the rebuttal, the authors conducted supplementary experiments to address these data integrity issues and extended their evaluation to a broader range of model scales, effectively resolving these concerns.
2. Practicality and Computational Efficiency (Reviewer wnXZ): In response to skepticism regarding the practical resource requirements of the proposed method, the authors provided a detailed runtime analysis. The results demonstrate that the framework yields tangible improvements in inference latency, successfully alleviating concerns about its deployment efficiency.

Outstanding Concerns:

3. Empirical Support for Theoretical Statements (Reviewers hGUM, wnXZ, hGc3): A primary remaining concern involves the authors' claim that richer reflections boost initial answer quality by exposing models to diverse problem-solving approaches. While the rebuttal proved that longer trajectories contain more varied ideas, reviewers noted that the causal link—whether these varied ideas directly improve the quality of the first attempt—remains insufficiently validated and lacks rigorous empirical evidence.
4. Performance vs. Efficiency Trade-off (Reviewer LMU1): While the proposed CAD and QRC methods successfully reduce token consumption, they incur non-trivial accuracy drops on challenging datasets. Reviewers remain concerned that this sacrifice in performance limits the methods' practical utility, as the accuracy decline across multiple benchmarks has not been sufficiently mitigated to justify the efficiency gains.

**Reviewer Scores:**

- viewers hGUM (6), wnXZ (4), and hGc3 (6): Scores are expected to remain unchanged. Despite the rebuttal, the concern regarding how "diverse reasoning paths" empirically boost initial answer quality remains unresolved, leaving the paper’s core claim insufficiently supported.
- Reviewer LMU1 (6): The score is likely to stay at 6. The reviewer remains skeptical of the method’s practical significance, as the non-trivial accuracy drops on hard sets offset the benefits of reduced token consumption.

---

### Decision · Program_Chairs · 2026-01-26

Reject